# Mind the State: Towards Unified, Context-Aware EEG-to-fMRI Synthesis

## Abstract

Functional magnetic resonance imaging (fMRI) provides dynamic measurements of human brain activity at high spatial resolution and depth, but its use is constrained by high cost, limited accessibility, and strict acquisition requirements. Synthesizing fMRI data from more accessible, non-invasive modalities such as electroencephalography (EEG) offers a promising alternative, enabling inference of deep brain activity from low-cost scalp recordings in naturalistic settings. Despite recent progress, existing EEG-to-fMRI translation methods typically require training separate models for individual brain regions and offer limited consideration of subject-level variability in brain dynamics. In this study, we propose **UniEFS**, a **uni**fied **E**EG-to-**f**MRI **S**ynthesis model that enables full-brain fMRI reconstruction while accommodating datasets with varying demographic and physiological contexts within a single model. **UniEFS** leverages a pretrained fMRI decoder to embed rich spatial priors, as well as condition-aware prompt tokens that encode subject-level and experimental metadata to handle heterogeneous datasets. We extensively evaluate our model performance on eyes-closed resting-state data and demonstrate that it can reliably reconstruct temporally-resolved whole-brain fMRI activity, with strong potential to generalize to task-based fMRI in a zero-shot setting.

## 1 Introduction

The ability to non-invasively monitor brain activity is essential for advancing both neuroscience research and clinical care. Electroencephalography (EEG) and functional magnetic resonance imaging (fMRI) represent two ends of the neuroimaging spectrum. EEG captures fast, millisecond-scale electrical signals from the scalp, offering a direct window into neural activity with excellent temporal resolution and broad accessibility (Nicolas-Alonso & Gomez-Gil, 2012; Tong & Thankor, 2009). However, it suffers from poor spatial resolution and limited sensitivity for mapping large-scale and deep-brain circuits (Cohen, 2017; Chang & Chen, 2021). In contrast, fMRI provides rich spatial detail by measuring blood oxygenation-level dependent (BOLD) signals driven by neurovascular coupling across the entire 3D volume of the brain (Logothetis, 2008; Matthews et al., 2006). Yet fMRI is expensive, infrastructure-intensive, and constrained by low temporal resolution. It is also largely inaccessible in under-resourced communities and outpatient settings, and may be contraindicated for patients with certain implants or conditions (Jalloul et al., 2023; van Beek et al., 2019; Geethanath & Vaughan Jr, 2019). These complementary characteristics raise an intriguing question: *Can we equip EEG with fMRI-like representational power?* If so, it would unlock a new paradigm for scalable, high-resolution brain monitoring and decoding using only a lightweight, real-time, and non-invasive sensor, transforming both clinical practice and cognitive neuroscience.

These factors motivate a growing interest in reconstructing fMRI signals from EEG, leveraging their underlying physiological correlation and the representational power of deep learning to bridge the spatial and temporal divide between these two modalities. A particularly underexplored area in this field involves the **eyes-closed, resting-state condition**. This condition is of significant interest in both research and clinical contexts: it offers a window into the brain's intrinsic functional organization and is widely used due to its simplicity and ease of implementation. It is especially valuable for its applicability to diverse populations, including children and patients who may not tolerate or comply with task-based paradigms. However, unlike task-based paradigms that provide clear temporal anchors, resting-state brain activity is more spontaneous and variable, spanning a variety of internal

brain states such as mind-wandering, vigilance fluctuations, or even light sleep, which makes it inherently more challenging to decode (Liu, 2016; Liu & Falahpour, 2020). For example, changes in vigilance introduce substantial non-stationarity: the decline from alertness into drowsiness and light sleep is accompanied by marked changes in the spectral content of EEG, and in the signal amplitude and network structure of fMRI (Liu & Falahpour, 2020; Martin et al., 2021).

Within the body of existing work, NeuroBOLT (Li et al., 2024b) is - to our knowledge - the only study to date that has explored EEG-fMRI translation under the eyes-closed resting-state condition. While demonstrating promising results with multi-dimensional EEG representation learning, it requires training separate models for individual brain regions, limiting efficiency and scalability. Another very recent approach, CATD (Yao et al., 2025), demonstrated efficient cortical surface fMRI generation by conditioning a diffusion model on EEG. However, by design, its reliance on fMRI surface maps restricts reconstruction to the cortex, leaving subcortical regions outside the model's representational space. Subcortical brain areas are increasingly recognized as vital to healthy cognition as well as a wide range of disease processes (Favaretto et al., 2022; Koshiyama et al., 2018; Shepherd, 2013). This work also included resting-state data, albeit during eyes-open conditions, which may not be as conducive to more dramatic shifts in vigilance (e.g., falling asleep) compared to eyes-closed conditions. Moreover, these approaches share a common limitation: they treat the EEG-fMRI relationship as largely uniform across individuals. This overlooks inter-subject variability driven by demographic factors (e.g., age and sex) and from dynamic, time-varying physiological states (e.g., drowsiness or vigilance), which are particularly pronounced during resting-state recordings. Such states have been shown to modulate both EEG and fMRI signals, as well as their correlations (Liu & Falahpour, 2020; Olbrich et al., 2009; Wong et al., 2013). Addressing this variability is therefore essential for developing scalable and generalizable EEG-to-fMRI translation frameworks that extend beyond specific conditions or cohorts. A more comprehensive review of related work is provided in Appendix A.

To address the issues discussed above, we propose a unified, context-aware framework for generalizable and efficient EEG-to-fMRI translation, operating in a frame-wise manner to reconstruct the fMRI frame at each time point. Rather than following prior work that relies solely on end-to-end training with scarce paired EEG–fMRI data, which may not adequately capture population-level variability, we adopt a two-stage strategy: **(1) uni-modal fMRI pretraining**, and **(2) cross-modal alignment**, where EEG signals are embedded to align with the learned fMRI latent space via a context-aware encoder. **Stage (1)** focuses on learning expressive and generalizable fMRI representations from larger-scale unpaired fMRI data with rich coverage of brain dynamics. A key motivation stems from the observation that fMRI activity exhibits structured spatial patterns even at the level of individual frames (Liu et al., 2018; 2013). In particular, co-activation pattern (CAP) analyses have revealed that groups of brain regions display recurring and instantaneous configurations of activation and deactivation (Liu et al., 2018). Building on this insight, and drawing inspiration from masked signal modeling (MSM) in vision, language, and neuroimaging (Chen et al., 2023; Xie et al., 2022; Radford et al., 2019; Yang et al., 2023; Jiang et al., 2024), we design a self-supervised masked modeling strategy that trains the model to recover masked brain regions from the visible context within each frame. This design encourages the model to capture transient spatial dependencies across regions and learn robust representations of instantaneous brain states. Given the domain shift between the pretraining corpus and the paired EEG-fMRI dataset, we then fine-tune the MAE on the fMRI portion of the EEG-fMRI dataset to obtain a domain-adapted encoder and decoder. In **Stage (2)**, we align EEG with this learned fMRI latent space and reconstruct fMRI with the pretrained decoder. To facilitate this, we introduce a context-aware EEG encoder that projects temporal and spectral features into the pretrained fMRI space, while explicitly incorporating auxiliary metadata. This contextual conditioning enables the model to account for individual variability in the EEG-fMRI relationship, thereby bridging the two modalities in a unified framework. We demonstrate that the resulting framework enables full-brain fMRI reconstruction from EEG within a unified model, without requiring region-specific supervision or subject-dependent customization. By leveraging uni-modal fMRI pretraining, domain adaptation, and latent alignment, UniEFS offers an effective and scalable solution for decoding intrinsic brain activity under eyes-closed resting-state conditions. The key contributions are summarized as follows:

**Context-aware EEG encoding.** To better accommodate heterogeneity in EEG-fMRI data (e.g., different acquisition sites, demographic attributes, and vigilance levels), we introduce prefix prompt tokens that encode dataset-specific and subject-level metadata, facilitating unified training across formats.

**Whole-brain EEG-to-fMRI synthesis.** We develop a unified framework that reconstructs fMRI activity, spanning hundreds of functional regions, from EEG using a single model. By first pretraining the fMRI decoder on unpaired fMRI data, we embed strong spatial priors to promote accurate and efficient reconstruction across all brain regions.

**Comprehensive evaluation of the predictive power.** We conduct extensive evaluations of fMRI time-series reconstruction performance across multiple brain areas, including cortical and subcortical regions, as well as whole-brain functional connectivity patterns. In addition, we evaluate the model's ability to generalize across experimental conditions, demonstrating strong zero-shot transfer performance and the model's broad predictive capacity.

## 2 METHODS

### 2.1 OVERVIEW

In this section, we describe the overall task setting and proposed framework of UniEFS. Our approach performs frame-by-frame fMRI prediction: given a sliding window of EEG signals preceding each fMRI time point, the model predicts the corresponding fMRI frame. This design enables flexible generation of fMRI sequences of arbitrary length. Our work mainly focuses on Regions-of-Interest-level (ROI-level) fMRI reconstruction, which offers a favorable trade-off between spatial resolution and efficiency. Compared to voxel-wise and surface-based methods, it reduces computational cost and improves signal-to-noise ratio (SNR), while also covering both cortical and subcortical regions for full-brain modeling. Moreover, as a representation adopted in recent fMRI foundation models (Dong et al., 2024; Caro et al., 2024; Thomas et al., 2022), ROI-level modeling provides a scalable and effective basis for future extensions.

However, achieving accurate frame-wise ROI-level reconstruction from EEG is non-trivial, due to the following key challenges. First, frame-wise reconstruction implicitly involves learning the projection from neuronal activity to its hemodynamic response, which is not uniform, varying across brain regions, individuals, and brain states. Second, paired EEG–fMRI datasets are scarce and moreover vary in their subject characteristics, potentially hindering generalization to broader populations and different conditions. To address these challenges, we propose a two-stage learning framework as illustrated in Figure 1: **(1) fMRI Pretraining and Adaptation via Masked Signal Modeling:** We first pretrain a powerful encoder-decoder model on unpaired fMRI datasets using a masked reconstruction objective. This stage enables the model to learn population-level representations of brain activity. To bridge the domain gap between pretraining and downstream application, we further fine-tune the pretrained model on the fMRI portion of the EEG-fMRI paired dataset, adapting the decoder to the target domain while preserving its generalization capacity. **(2) Context-aware EEG-to-fMRI Mapping:** In the second stage, we integrate a dedicated EEG encoder, conditioned on demographic and physiological priors, with the adapted fMRI decoder. The EEG encoder learns to map temporal and spectral features of EEG signals into the corresponding fMRI latent space, enabling full-brain fMRI reconstruction.

### 2.2 STAGE 1: FMRI PRETRAINING AND ADAPTATION VIA MASKED SIGNAL MODELING (F-MSM)

**Pretraining.** Functional MRI measures brain activity via blood-oxygen-level-dependent (BOLD) signals represented as 3D volumes over time. To reduce dimensionality and improve signal-to-noise ratio (SNR), these signals are commonly summarized using brain parcellation techniques, which average voxel-wise signals within predefined regions of interest (ROIs), forming a 1D ROI vector per time point. This yields a parcellated fMRI matrix denoted as $Y \in \mathbb{R}^{P \times K}$, where $P$ is the number of ROIs and $K$ is the total number of time points. Here, we employ the Dictionaries of Functional Modes (DiFuMo) parcellation (Dadi et al., 2020) with P=512, which provides fine-grained, whole-brain coverage. During pretraining, each 1D ROI vector corresponding to a single time point is treated as an individual training sample, yielding $K$ samples per fMRI scan. Although parcellation reduces voxel-level redundancy, functional dependencies and spatial correlations persist across brain regions due to the network-level organization of brain activity. To encourage the model to capture these intrinsic patterns, for each of the above ROI vectors, we adopt a high masking ratio (50%) during pretraining, forcing the network to infer random missing regional signals from the surrounding context. This

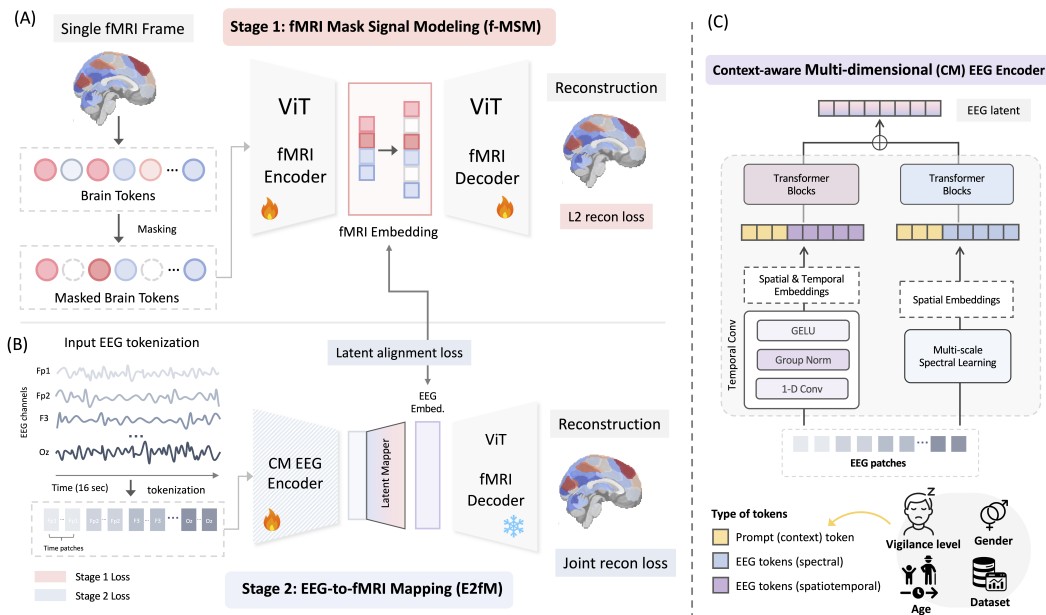

Figure 1: **Overall framework.** (A) Stage 1: Masked signal modeling on fMRI frames. (B) Stage 2: EEG-to-fMRI mapping. The pretrained fMRI encoder and decoder are frozen in this stage. (C) Context-aware EEG embedding.

design promotes the learning of expressive, population-level representations that generalize across individuals and tasks. For this reconstruction task, we employ a transformer-based architecture (Chen et al., 2023; 2024; Dosovitskiy et al., 2020), where each ROI is treated as an individual token. An encoder processes only the visible (unmasked) ROIs, and a lightweight decoder is trained to reconstruct the complete ROI vector based on the contextual information inferred from the unmasked regions.

**Fine-tuning.** After pretraining on public fMRI datasets, we adapt the model using the fMRI portion of our EEG–fMRI paired training set, with each scan preprocessed into ROI-level time series under the same parcellation scheme as used in pretraining.

**Training Objective.** Following He et al. (2022), the reconstruction loss (MSE) is computed solely on the masked tokens for both pretraining and finetuning.

### 2.3 STAGE 2: CONTEXT-AWARE EEG-TO-FMRI MAPPING

Following Li et al. (2024b), we extract the EEG window spanning a duration $T$, corresponding to the approximate latency of the hemodynamic response function (HRF), before each fMRI frame collection. This forms an EEG-fMRI paired input-output sample denoted as $\{X, Y^{(\text{paired})}\}$, where $X \in \mathbb{R}^{C \times T}$ represents the multichannel EEG input with $C$ channels and $T$ time points, and $Y^{(\text{paired})} \in \mathbb{R}^P$ denotes the corresponding parcelled fMRI ROI vector with $P$ ROIs. The EEG input window $X$ is first processed by the EEG encoder $\mathcal{E}_{\text{EEG}}$ to generate a latent representation, which is then passed to the pretrained domain-adapted decoder $\mathcal{D}_{\text{fMRI}}$ obtained from f-MSM. Overall, given the full model $f_\theta(.)$, the overall fMRI reconstruction task can be formulated as $\hat{Y}_t^{(\text{paired})} = f_\theta(X_{t-T:t-1})$, where $\hat{Y}_t^{(\text{paired})} \in \mathbb{R}^P$ is the reconstructed fMRI frame at time index $t$.

**EEG Encoder.** Our objective is to enable EEG-driven fMRI reconstruction by aligning EEG representations with the fMRI latent embedding space. To achieve this, we adapt the multi-dimensional encoder from NeuroBOLT (Li et al., 2024b) as the backbone encoder, a transformer-based architecture designed to capture rich and complementary spatial, temporal, and multi-scale spectral information from EEG signals. We first segment a EEG window $X$ into non-overlapping patches using a window of length $w$, yielding a sequence of patches $x_{c,k} \in \mathbb{R}^w$ for each channel $c = 1, \ldots, C$ and patch index

$k = 1, \ldots, \lfloor T/w \rfloor$. These patches are then fed into (i) spatiotemporal module (a pretrained EEG encoder adapted from the EEG foundation model LaBraM (Jiang et al., 2024)) and (ii) multi-scale spectral transformer modules to generate two EEG latent embeddings $\mathbf{z}_{\text{EEG}_{st}}, \mathbf{z}_{\text{EEG}_{sp}} \in \mathbb{R}^{(C \times \frac{T}{w}) \times D}$ respectively, where $D$ is the embedding dimension. Instead of applying global average pooling across the token dimension (Li et al., 2024b; Jiang et al., 2024; Yang et al., 2023), we retain the full sequence of token embeddings to preserve fine-grained spatial and temporal information. The embeddings from two modules are summed up and then passed through a latent mapping module, which consists of two linear projections to align the EEG embeddings with the dimensionality and structure of the fMRI latent space. This final latent representation is then passed to the fine-tuned decoder $\mathcal{D}$ to reconstruct the full-brain fMRI signal.

**Prefix Prompt Injection.** To incorporate auxiliary information (Gao et al., 2024) and enhance the generalization of EEG representations, we introduce a set of learnable prefix prompts in the EEG encoder that are concatenated to the EEG patches prior to the transformer modules. These prompt tokens are designed to encode subject- and dataset-specific metadata and are optimized jointly with the rest of the model, enabling the network to adaptively condition its representation based on contextual information. Specifically, we include the following prompt tokens: **(1) Dataset tokens**: learnable embeddings of shape $\mathbb{R}^{J \times D}$, where $J$ is a tunable hyperparameter, representing the number of dataset tokens ($J = 5$ in our experiments). Each dataset is assigned its own set of tokens, enabling the model to capture dataset-specific characteristics. **(2) Age token**: A single token generated by projecting the subject's age through a linear embedding layer. **(3) Sex token**: A learnable token indicating the subject's biological sex (e.g., male or female). **(4) Vigilance token**: A categorical learnable token encoding the vigilance level (drowsy, intermediate, alert) at each fMRI frame (TR), allowing the model to condition its reconstruction on frame-specific vigilance states. All prefix tokens share the same embedding dimension $D$ as the EEG patch embeddings output by the EEG encoder. Let $N_{\text{prompt}}$ be the total number of prompt tokens used (e.g., $N_{\text{prompt}} = 8$ when all components are included), such that $\mathbf{z}_{\text{prompt}} \in \mathbb{R}^{N_{\text{prompt}} \times D}$ represents all prompt tokens. These tokens are then concatenated to the EEG embedding $\mathbf{z}_{\text{EEG}}$ along the token dimension to form the augmented token sequence $\mathbf{z} \in \mathbb{R}^{(N_{\text{prompt}} + N_{\text{EEG}}) \times D}$. This enriched EEG representation is then passed to the following transformers and to the alignment module for EEG-to-fMRI mapping.

**EEG-fMRI Embedding Alignment.** To align the EEG embeddings with the fMRI latent space, during training, we first obtain a reference embedding from fMRI by passing ground-truth fMRI vector $Y^{(\text{paired})}$ through the frozen fine-tuned fMRI encoder $\mathcal{E}_{\text{fMRI}}$. This yields the fMRI latent embedding:

$$\mathbf{E}_{\text{fMRI}} = \mathcal{E}_{\text{fMRI}}(Y^{(\text{paired})}) \in \mathbb{R}^{P \times D_{\text{fMRI}}}, \tag{1}$$

where $D_{\text{fMRI}}$ is the dimension of the fMRI latent space.

In parallel, the EEG input $X$ is first processed by the EEG encoder $\mathcal{E}_{\text{EEG}}$ to obtain patch-wise latent embeddings $\mathbf{z}$. To align this representation with the fMRI latent space, we apply a linear projection module $\mathcal{P}$ to map the enriched EEG embedding into the same shape as the fMRI embedding:

$$\mathbf{E}_{\text{EEG}}^{\text{proj}} = \mathcal{P}(\mathbf{E}_{\text{EEG}}) \in \mathbb{R}^{P \times D_{\text{fMRI}}}, \tag{2}$$

which is then used in the alignment and decoding modules.

To extract compact and semantically aligned representations between $\mathbf{E}_{\text{fMRI}}$ and $\mathbf{E}_{\text{EEG}}^{\text{proj}}$, we adopt a LoRA-inspired low-rank projection mechanism (Hu et al., 2022). Specifically, we define two learnable matrices: $\mathbf{B} \in \mathbb{R}^{D_{\text{fMRI}} \times r}$ and $\mathbf{A} \in \mathbb{R}^{r \times D_{\text{comp}}}$, where $r$ is the intermediate rank and $D_{\text{comp}}$ is the final compressed dimension. The final low-rank embeddings are computed as:

$$\tilde{\mathbf{E}}_{\text{fMRI}} = \mathbf{E}_{\text{fMRI}} \cdot \mathbf{B} \cdot \mathbf{A} \in \mathbb{R}^{P \times D_{\text{comp}}}, \quad \tilde{\mathbf{E}}_{\text{EEG}} = \mathbf{E}_{\text{EEG}}^{\text{proj}} \cdot \mathbf{B} \cdot \mathbf{A} \in \mathbb{R}^{P \times D_{\text{comp}}}. \tag{3}$$

To encourage alignment between the low-rank EEG and fMRI embeddings, we optimize the mean squared error (MSE) between the two latent embeddings:

$$\mathcal{L}_{\text{align}} = \frac{1}{P} \sum_{p=1}^{P} \left| \tilde{\mathbf{E}}_{\text{EEG}}^{(p)} - \tilde{\mathbf{E}}_{\text{fMRI}}^{(p)} \right|_2^2, \tag{4}$$

where $\tilde{\mathbf{E}}^{(p)}$ is the embedding for the $p$-th ROI.

**Training Objective.** Our framework is trained using a joint objective that combines **(1) a latent alignment loss**, which uses MSE to align EEG and fMRI representations in latent space, and **(2) a reconstruction loss**, which ensures accurate full-brain fMRI prediction by comparing the decoder's output to the ground truth fMRI vector. The **reconstruction loss** is formulated as a weighted combination of MSE and spatial correlation (SCorr) to capture both absolute and relative accuracy:

$$\mathcal{L}_{\text{recon}} = \lambda_{\text{MSE}} \cdot \mathcal{L}_{\text{MSE}} + (1 - \lambda_{\text{MSE}}) \cdot \mathcal{L}_{\text{SCorr}}. \tag{5}$$

Here, the SCorr term is defined as:

$$\mathcal{L}_{\text{SCorr}} = 1 - \text{Corr}(\hat{Y}^{(\text{paired})}, Y^{(\text{paired})}), \tag{6}$$

where $\text{Corr}(\hat{Y}^{(\text{paired})}, Y^{(\text{paired})})$ denotes the Pearson correlation coefficient between the predicted fMRI ROI vector and the ground-truth vector. This term encourages the model to match not only the absolute magnitudes but also the *relative spatial pattern* of regional activations within each time point.

The overall training objective of Stage 2 is a weighted sum of the alignment and reconstruction losses:

$$\mathcal{L}_{\text{total}} = \lambda_{\text{align}} \cdot \mathcal{L}_{\text{align}} + \lambda_{\text{recon}} \cdot \mathcal{L}_{\text{recon}}, \tag{7}$$

where $\lambda_{\text{align}}$ and $\lambda_{\text{recon}}$ are hyperparameters that balance the contribution of each loss term.

## 3 EXPERIMENTS

### 3.1 DATASETS AND PREPROCESSING

**Pretraining fMRI Dataset.** A subsample of resting-state fMRI (rs-fMRI) data from the HCP 1200-subject release (Van Essen et al., 2012) was used for pretraining. Subjects were scanned up to 4 times, twice on one day and twice on a second day. We included only those subjects who completed all four runs and were reported to have passed quality control in Xifra-Porxas et al. (2021); Power et al. (2017), resulting in 375 subjects (n = 1500 scans). The rs-fMRI scans in this dataset were acquired with a temporal resolution (TR) of 0.72 seconds, a duration of 1,200 frames per run (14.4 minutes), and a spatial resolution of 2 mm isotropic. Please refer to the Appendix B for preprocessing details.

**Resting-state Simultaneous EEG-fMRI Datasets.** **Dataset 1** is a shared dataset from Li et al. (2024b). This dataset comprises 29 simultaneous EEG-fMRI scans from 22 healthy subjects, with 7 participants having two scans. Each scan lasts 20 minutes (575 TR, TR = 2.1 seconds). Scalp EEG was recorded using a 32-channel MR-compatible system (10–20 layout). Additional details on data acquisition and preprocessing can be found in Li et al. (2024b). **Dataset 2** is a different in-house rs-EEG-fMRI dataset. It comprises 10 scans from 7 healthy participants, with 3 individuals undergoing two scans each. During the scans, participants rested passively with their eyes closed. Written informed consent was obtained from all participants, and all procedures were approved by the Institutional Review Board. BOLD fMRI data were acquired on a 3T Siemens Prisma scanner using a multi-echo gradient-echo EPI sequence (TR = 2.1 seconds). Simultaneous scalp EEG was recorded using a 32-channel MR-compatible system (10–20 layout). To ensure consistency, this dataset was preprocessed using the same pipeline as Dataset 1. Further details on data collection and preprocessing are provided in Appendix B.

**Vigilance Score.** The vigilance state is a categorical score with three classes (drowsy, intermediate, and alert) assigned to each fMRI frame. This classification is derived from EEG data based on Vigilance Algorithm Leipzig (VIGALL 2.1 add-on for Brain Vision Analyzer [1]) (Olbrich et al., 2015; Huang et al., 2015; Jawinski et al., 2019). VIGALL stages each 1-second segment of preprocessed EEG into one of five vigilance levels (A1, A2, A3, B1, B2/3), reflecting decreasing levels of arousal. These vigilance stage labels are then grouped into 63-second epochs (corresponding to 30 fMRI time points), and the distribution of stages within each epoch is used to assign a single vigilance class, i.e., alert, intermediate, or drowsy, to that epoch. This final label is then propagated to each of the 30 fMRI frames within the corresponding epoch. Note that vigilance labels were shifted forward by 5 seconds (~2 TRs) to account for the temporal delay between neural activity and the peak BOLD response. Please see the Appendix B for further details.

---

[1] https://brainvision.com/products/analyzer-2/

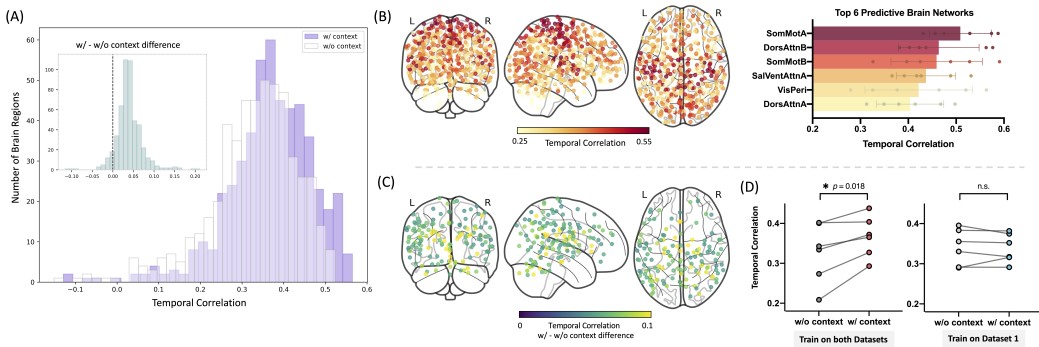

Figure 2: **Reconstruction performance.** (A) Distribution of average prediction performance for all brain regions with (w/) and without (w/o) context embedding. (B) Visualization of reconstruction performance and top predictive brain networks; dots represent brain regions. (C) Regional differences in performance. (D) Sample-wise differences in performance; points represent test scans.

## 3.2 EXPERIMENTAL SETUP

**Data Preparation.** From the preprocessed fMRI data in the paired and unpaired datasets, we extract time courses from regions of interest using the DiFuMo atlas (Dadi et al., 2020) with P=512 for full brain coverage. We regress out six motion-related confounds, apply a low-pass filter with a cutoff at 0.15 Hz (which captures the low-frequency band typically of interest in rs-fMRI studies), and z-normalize each ROI time series. For EEG preprocessing, we remove non-EEG channels, including ECG, EOG, and EMG, retaining 26 channels in Dataset 1 and 31 channels in Dataset 2. We select the **23** common EEG channels from these datasets for joint training. The EEG data are resampled to 200 Hz to improve computational efficiency while preserving relevant frequency information. For each fMRI time point, we extract a 16-second EEG window preceding the scan, strictly following the protocol described in Li et al. (2024b). EEG amplitude normalization is also performed, where the signal is divided by 100 to ensure that the majority of values fall within the range of -1 to 1.

**Baselines.** We compare our model with three open-source EEG-to-fMRI synthesis baselines (Kovalev et al., 2022; Li et al., 2024b;a) and state-of-the-art EEG encoders including recent foundation models (please refer to Appendix C.2 for details). These baseline models were originally designed or benchmarked in Li et al. (2024b) under a region-specific setting. To enable comparison on multi-region reconstruction, we adapt them by modifying the final projection layer to map the latent embeddings to the entire set of ROIs, thus extending them into multi-region baselines.

**Implementation Details.** All experiments are conducted on a single NVIDIA RTX A5000 GPU using Python 3.9.12, PyTorch 2.0.0, and CUDA 11.8. The training set for stage 1 consists of 1,200 scans, with 300 scans used for validation, resulting in approximately 720,000 training samples (one per time point). During stage 2, we train the model to predict fMRI signals across entire unseen scans using EEG, and use the same data partitioning strategy as in Li et al. (2024b) (an approximately 3:1:1 split for unseen-subject whole-scan reconstruction). We incorporate Dataset 2 as additional training data, resulting in a total of 28 training scans, 5 validation scans, and 6 test scans. Scans from the same individual are always assigned to the same split (training, validation, or test), with train/validation/test subjects strictly disjoint, since data from the same subject may have similar latent representations. For reproducibility, a fixed random seed is used across all experiments. Please refer to Appendix C for optimization and training details.

## 3.3 MAIN RESULTS

Our model was trained to predict held-out recordings across entire 20-minute scans using 23 EEG electrodes. We compare UniEFS with state-of-the-art EEG-to-fMRI translation baselines (Table 1) and EEG encoders (Table 5 in Appendix D.1), finding that UniEFS consistently outperforms the others in reconstructing regional time courses and has the second-best performance in recovering FC. We refer readers to Appendix E.1 for a detailed discussion on potential factors contributing to this observation and D.15 for visualization examples of reconstructed time series.

Table 1: Comparison of different models on full brain fMRI reconstruction. MM: Whether the model is originally designed for multi-region prediction; FB: Full brain; GM: Cortical gray matter; SC: Sub-cortical regions; CB: Cerebellum; Conn: Metric is applied on the full brain functional connectivity (FC) matrix; TCorr: Temporal correlation between prediction and ground truth; PCorr: Pixel-wise correlation between predicted and real FCs. **Bold**: the best; Underlined: the second best. Values are shown as mean ± std. Paired t-test significance between our model and each baseline is shown using color codes: blue (p<0.05), yellow (p<0.01), red (p<0.001), uncorrected.

| Model Name | MM | FB TCorr ↑ | GM TCorr ↑ | SC TCorr ↑ | CB TCorr ↑ | Conn PCorr ↑ | Conn MSE ↓ |
|---|---|---|---|---|---|---|---|
| Ours | ✓ | **0.367 ± 0.052** | **0.394 ± 0.060** | **0.276 ± 0.082** | **0.247 ± 0.060** | 0.527 ± 0.084 | 0.233 ± 0.072 |
| NeuroBOLT (Li et al., 2024b) | ✗ | 0.331 ± 0.044 | 0.357 ± 0.049 | 0.258 ± 0.092 | 0.216 ± 0.046 | 0.455 ± 0.079 | 0.349 ± 0.097 |
| Li et al. (Li et al., 2024a) | ✗ | 0.312 ± 0.038 | 0.329 ± 0.037 | 0.253 ± 0.090 | 0.236 ± 0.058 | **0.535 ± 0.077** | **0.217 ± 0.065** |
| BEIRA (Kovalev et al., 2022) | ✗ | 0.171 ± 0.148 | 0.196 ± 0.170 | 0.086 ± 0.085 | 0.063 ± 0.073 | 0.459 ± 0.080 | 0.368 ± 0.090 |

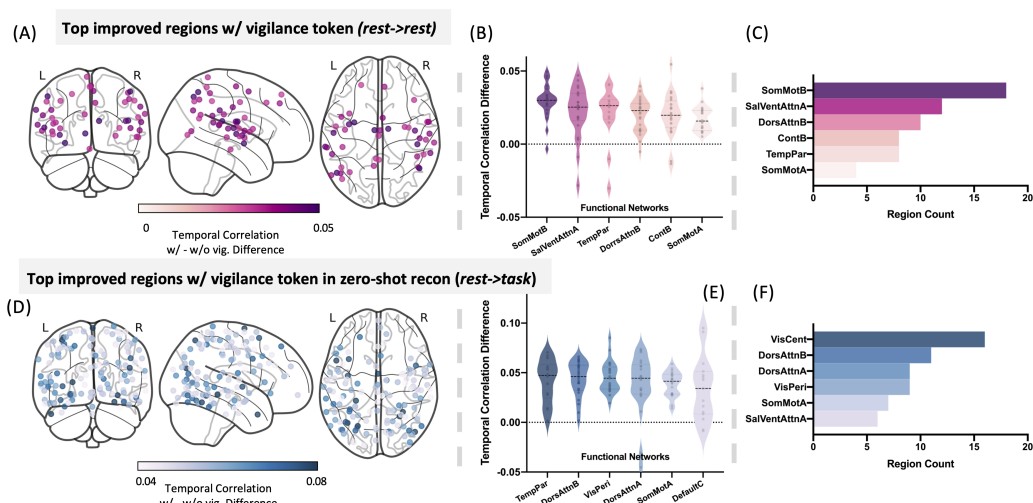

Figure 3: **Performance improvements with vigilance embedding.** (A-C) Improvements when train and test on resting-state condition. (D-F) Improvements when train on resting-state and test on task condition (zero-shot) (A, D) Brain regions showing the greatest performance improvement. (B, E) Distribution of region-wise improvement within brain networks. (C, F) The top 100 regions benefitting from vigilance embedding were selected and assigned to their corresponding brain networks. The 6 networks with the highest counts among these top-ranked regions is shown.

Figure 2(A-C) presents a comprehensive evaluation of model performance, including region-wise distributions of predictive accuracy across brain areas with and without context embeddings. The results highlight the effectiveness of incorporating context embeddings, and indicate that activity in the somatomotor network is most reliably predicted from EEG signals, followed by the dorsal attention and salience/ventral attention networks. We further investigate the effectiveness of context embeddings under varying amounts of training data. As shown in Figure 2(D), when training on both datasets, the inclusion of context embeddings significantly improves prediction performance across individual scans (paired t-test, $p = 0.018$). However, when training is limited to Dataset 1 alone and evaluated on the same test scans, the performance difference between models with and without context embedding is negligible. This suggests that the benefit of incorporating context information emerges only when the training data exhibit sufficient variability, such as differences in subjects, vigilance levels, or population characteristics, allowing the model to meaningfully leverage auxiliary metadata.

To better understand the role of context information, we further analyze the effect of incorporating vigilance embeddings. Specifically, we compare model performance with and without vigilance conditioning to identify brain regions that benefit the most from this additional context. As shown in Figure 3, several regions, particularly within sensory-motor network, salience and attention-related networks, as well as one thalamus region, show marked improvement when vigilance information is included. These regions have been consistently reported in the fMRI literature as being associated

with vigilance, and the spatial distribution observed in Figure 3(A) closely overlaps with vigilance-related fMRI maps reported by previous studies (Liu & Falahpour, 2020; Schneider et al., 2016; Goodale et al., 2021). Under zero-shot resting-state to auditory-task generation setting (Figure 3 (D-F)), we found Temporo-Parietal (TempPar) Network and Dorsal Attention Network (DAN) show the most pronounced improvements. Specifically, TempPar is known to support attentional reorienting, sensory–motor integration, and response preparation - functions that are strongly modulated by moment-to-moment vigilance and arousal. In the auditory task dataset, participants must detect auditory cues and make rapid button-press responses; thus, fluctuations in alertness directly impact both auditory processing efficiency and motor readiness, processes for which TempPar plays a central role. Likewise, DAN is among the networks most sensitive to arousal and sustained attention. These results support the biological plausibility of our approach and provide compelling evidence that incorporating vigilance context enables more accurate and interpretable EEG-to-fMRI translation, particularly in regions sensitive to fluctuations in arousal and attention, which is especially crucial to consider for resting-state data.

### 3.4 GENERALIZATION PERFORMANCE AND ABLATION STUDIES

**Generalization.** We evaluate the zero-shot performance of our multi-region model on an unseen auditory task-based dataset from Li et al. (2024b) (Figure 4). Our model generalizes well to task-induced fMRI dynamics, capturing prominent brain activity features despite not being trained on task-based data. A comprehensive analysis of various rest-to-task transfer strategies, including fine-tuning, joint training and personalized-finetuning, is provided in Appendix D.2. To further validate the quality of the zero-shot generated fMRI and its ability to reflect true subject-specific patterns in FC, we additionally performed a connectome fingerprinting analysis, as described in Appendix D.3. Notably, our model's zero-shot predicted fMRI also demonstrated high fingerprinting accuracy across full-brain, gray matter, and subcortical FC matrices (see Table 6 in Appendix). These findings suggest that the generated fMRI signals preserve individualized FC signatures, supporting their potential utility in downstream applications involving subject-specific brain representations, such as cognitive trait identification, behavioral decoding, and clinical profiling (Finn et al., 2015; Mantwill et al., 2022; Lu et al., 2024).

Table 2: Ablation study on model components. Each row removes one input or loss function from the full model. vig.: vigilance token; demo.: demographic token; d.t.: dataset token; w/o: without. Paired t-test significance between our model and each ablation is shown using color codes: blue ($p<0.05$), yellow ($p<0.01$), red ($p<0.001$), uncorrected.

| Model Type | FB TCorr | GM TCorr | Conn PCorr | Conn MSE |
|---|---|---|---|---|
| Full (context + 5 d.t.) | **0.367 ± 0.052** | **0.394 ± 0.060** | 0.527 ± 0.084 | **0.233 ± 0.072** |
| w/o fine-tune | 0.365 ± 0.056 | 0.391 ± 0.064 | 0.527 ± 0.066 | 0.267 ± 0.075 |
| w/o pretrain | 0.329 ± 0.033 | 0.361 ± 0.039 | 0.315 ± 0.074 | 0.439 ± 0.102 |
| w/o vig. | 0.357 ± 0.040 | 0.381 ± 0.044 | **0.532 ± 0.076** | 0.237 ± 0.070 |
| w/o demo. | 0.351 ± 0.060 | 0.378 ± 0.068 | 0.503 ± 0.081 | 0.271 ± 0.082 |
| w/o demo. & vig. | 0.327 ± 0.075 | 0.356 ± 0.089 | 0.491 ± 0.079 | 0.288 ± 0.089 |
| w/o age | 0.351 ± 0.047 | 0.376 ± 0.052 | 0.513 ± 0.088 | 0.261 ± 0.078 |
| w/o sex | 0.358 ± 0.041 | 0.382 ± 0.044 | 0.529 ± 0.079 | 0.235 ± 0.071 |
| w/o d.t. | 0.355 ± 0.049 | 0.380 ± 0.055 | 0.527 ± 0.081 | 0.234 ± 0.069 |
| 1 d.t. | 0.349 ± 0.051 | 0.375 ± 0.059 | 0.494 ± 0.084 | 0.288 ± 0.085 |
| 10 d.t. | 0.356 ± 0.048 | 0.381 ± 0.054 | 0.527 ± 0.081 | 0.236 ± 0.070 |
| w/o $\mathcal{L}_{\text{align}}$ | 0.339 ± 0.052 | 0.367 ± 0.055 | 0.502 ± 0.082 | 0.280 ± 0.083 |
| w/o $\mathcal{L}_{\text{recon}}$ | 0.147 ± 0.022 | 0.150 ± 0.025 | 0.172 ± 0.052 | 0.266 ± 0.042 |
| w/o $\mathcal{L}_{\text{SCorr}}$ | 0.355 ± 0.049 | 0.384 ± 0.056 | 0.500 ± 0.083 | 0.293 ± 0.088 |
| w/o $\mathcal{L}_{\text{MSE}}$ | 0.347 ± 0.050 | 0.376 ± 0.058 | 0.481 ± 0.085 | 0.317 ± 0.092 |

**Ablation Studies.** We evaluate the contribution of each component (Table 2). We first compared our full model against a variant in which the transformer-based fMRI decoder was trained from scratch, without pretraining. Results indicate that without pretraining, the model struggles to capture temporal correlations between brain regions, leading to significantly reduced performance. Next, we examined the impact of different components of the context prompt. We found that removing any of the context tokens resulted in a drop in performance, highlighting their importance. The number of dataset tokens serves as a tunable hyperparameter. We found that using five tokens was sufficient to effectively handle the two training datasets used in our experiments. Beyond these ablations, we also

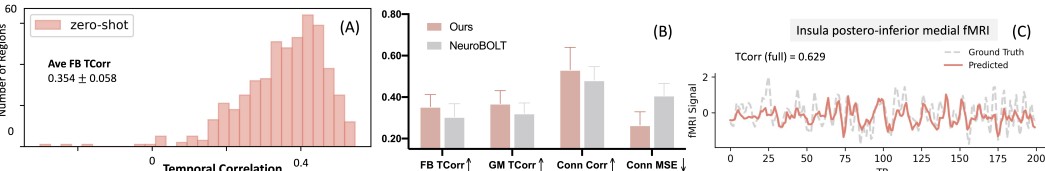

Figure 4: **Zero-shot reconstruction on task-condition data.** (A) Prediction performance distribution for all 512 regions. (B) Performance comparison with the baseline. (C) Example of reconstructed time series within the insula, shown for part of a scan.

analyzed the impact of data scale (Figure 10), mask ratio (Table 8), and patch size (Table 9) during fMRI pretraining, please see Appendix D.8, D.9, D.10 for details.

## 4 CONCLUSION

We introduce **UniEFS**, a unified framework for reconstructing full-brain fMRI activity from EEG. By leveraging large-scale fMRI pretraining, followed by domain adaptation and cross-modal alignment, our model effectively bridges the spatial-temporal gap between EEG and fMRI without relying on region-specific supervision or subject-dependent customization. We incorporate context-aware EEG encoding using metadata-informed prompt tokens, enabling the model to account for physiological and demographic variability that modulates EEG-fMRI correspondence. Our results demonstrate that UniEFS achieves state-of-the-art performance in time-resolved fMRI signal reconstruction across hundreds of brain regions, and its potential to recover functional connectivity. Our results highlight the effectiveness of combining self-supervised fMRI representation learning with context-conditioned EEG encoding for generalizable, scalable, context-aware, and interpretable cross-modality translation. UniEFS paves the way for real-world applications where high-resolution, fMRI-like insights could be derived from lightweight, portable EEG systems, enabling more accessible neuroimaging in clinical, cognitive, and mobile settings.

## ETHICS STATEMENT

All study procedures were conducted with approval from the Institutional Review Board (IRB), and the research is classified as posing minimal risk to participants since both EEG and fMRI are non-invasive neuroimaging modalities. Individuals with contraindications to MRI were precluded from study participation. During our data collection, participants rested passively with their eyes closed and received compensation for their participation. Written informed consent was obtained from all participants prior to enrollment. The protocols explicitly outlined the nature of the resting-state scanning, any possible discomfort, and the participants' right to withdraw at any time without penalty. We confirm that no vulnerable populations were targeted or exploited, and the study did not involve any experimental manipulations beyond routine neuroimaging procedures. Collected data were anonymized and handled in compliance with appropriate privacy and confidentiality safeguards to minimize the risk of re-identification or misuse.

## REPRODUCIBILITY STATEMENT

To ensure reproducibility, we have made our experimental setup as transparent and accessible as possible. Key model architectures, training protocols, and evaluation metrics are described in detail in Sections 2 and 3 of the main text. Additional implementation details and hyperparameter settings are provided in the Appendix B and C. Upon acceptance, we will release our complete source code, pretrained model weights, and the in-house EEG-fMRI dataset (i.e., Dataset 2) used in this study under an appropriate data-sharing agreement. This will allow the community to fully reproduce and build upon our results.

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

## A    RELATED WORK

EEG-to-fMRI reconstruction, while currently an underexplored research area, has received growing attention in recent years owing to advances in deep learning and cross-modality synthesis. Several studies (Liu & Sajda, 2023a;b; Calhas & Henriques, 2022; Lanzino et al., 2024) have proposed methods for reconstructing volume-wise fMRI spatial patterns from EEG signals. However, these approaches generally lack quantitative evaluation of temporal dynamics, that is, how brain activity evolves over time, or of temporal correlation across brain regions, which are critical readouts, under-pinning analyses of functional connectivity, network dynamics, and brain state transitions. Without assessing these aspects, it remains unclear whether such models can support broader neuroscientific or clinical applications that rely on accurate reconstruction of brain-wide temporal structure. A complementary line of work has investigated fMRI time series reconstruction in specific brain regions, particularly in subcortical regions, such as early work using ridge regression (Meir-Hasson et al., 2014; Or-Borichev et al., 2023), along with more recent deep-learning studies using seq-to-seq models (Kovalev et al., 2022; Li et al., 2024a). Yet, these models are typically trained on a within-subject basis, which hinders generalizability to new individuals. Furthermore, the majority of EEG-fMRI synthesis efforts have been limited to task-based paradigms, where external cues help structure the neural responses (Kovalev et al., 2022; Li et al., 2024a; Liu & Sajda, 2023a;b; Wei et al., 2020). As a result, spontaneous resting-state activity, particularly in the natural eyes-closed condition, remains largely unexplored. To bridge this gap, recent work Li et al. (2024b) introduced a transformer-based framework for reconstructing fMRI time series in a few selected brain regions during eye-closed resting-state. While this method shows promising generalization, it still requires training separate models for each target region, limiting scalability and efficiency. A more recent work by Yao et al. (2025) demonstrated efficient cortical surface fMRI generation by conditioning a diffusion model on EEG. However, its reliance on fMRI surface maps restricts reconstruction to the cortex, leaving subcortical regions outside the model's representational space.

## B    DATASET AND PREPROCESSING

**Pretraining fMRI Dataset: Preprocessing**    The HCP dataset (here, used for pretraining) had been processed using the HCP Minimal Preprocessing Pipeline (Glasser et al., 2013). In addition to this standard preprocessing, we removed low-order trends (polynomials up to the 4th order) to mitigate scanner drift artifacts, and temporally downsampled the data by a factor of 2, resulting in a final temporal resolution of 1.44 seconds and 600 frames per scan. This step was performed to render the temporal resolution more comparable to conventional fMRI scans, including those of the EEG-fMRI datasets used in our study. After extracting time courses from regions of interest using DiFuMo atlas (Dadi et al., 2020), we additionally regress out six rigid-body head-motion parameters (translation and rotation), apply a low-pass filter with a cutoff at 0.15 Hz (which captures the low-frequency band typically of interest in rs-fMRI studies), and z-normalize each ROI time series.

**Details about paired EEG-fMRI Dataset 2**    Dataset2 is an in-house resting-state EEG-fMRI dataset. It comprises 10 scans from 7 healthy participants, with 3 individuals undergoing two scans each. During the scans, participants rested passively with their eyes closed. Written informed consent was obtained from all participants, and all procedures were approved by the Institutional Review Board. MRI data were acquired on a 3T Siemens Prisma scanner. The T1-weighted structural images were collected with the following parameters: TR = 2200 ms, TE = 4.25 ms, flip angle = 9 deg, 1 mm isotropic. BOLD fMRI images were collected using multi-echo gradient-echo EPI sequence with TR = 2100 ms, echo times = 13.0, 29.4, and 45.7 ms, voxel size = 3 × 3 × 3 mm³, slice gap = 1 mm, matrix size = 82 × 50, 30 axial slices. MRI scanner triggers were recorded together with the EEG signals for data synchronization. The first seven volumes in fMRI data were dropped to allow magnetization to reach steady state. The fMRI preprocessing steps are kept consistent with Li et al. (2024b). Specifically, the steps included slice-timing and motion coregistration, noise reduction using multi-echo ICA which is implemented in tedana 0.0.9a[2], alignment to an MNI152 standard template (matrix shape: 91 × 109 x 91), removal of low-order trends (up to 4th-order polynomials), and spatial smoothing (to 3mm FWHM) using AFNI [3]. Simultaneous scalp EEG was acquired using

---

[2]https://tedana.readthedocs.io/en/stable/

[3]https://afni.nimh.nih.gov/afni

a 32-channel MR-compatible system (10–20 layout, FCz reference; BrainAmps MR, Brain Products GmbH) at a sampling rate of 5 kHz. The EEG system was synchronized to the scanner's 10 MHz clock to support gradient artifact correction. Preprocessing included removal of MR-related artifacts using BrainVision Analyzer 2 (Brain Products, Munich, Germany) (Moehlman et al., 2019; Allen et al., 2000), followed by downsampling to 250 Hz. No additional filtering was applied. The full set of 32 EEG channel labels is: ['FP1', 'FP2', 'F3', 'F4', 'C3', 'C4', 'P3', 'P4', 'O1', 'O2', 'F7', 'F8', 'T7', 'T8', 'P7', 'P8', 'FZ', 'CZ', 'PZ', 'OZ', 'FC1', 'FC2', 'CP1', 'CP2', 'FC5', 'FC6', 'CP5', 'CP6', 'TP9', 'TP10', 'POZ']. For joint training across Dataset 1 (Li et al., 2024b) and Dataset 2, we used the intersection of their channel sets, resulting in 23 overlapping channels: ['FP1', 'FP2', 'F3', 'F4', 'C3', 'C4', 'P3', 'P4', 'O1', 'O2', 'F7', 'F8', 'T7', 'T8', 'P7', 'P8', 'FZ', 'CZ', 'PZ', 'OZ', 'TP9', 'TP10', 'POZ'].

**Auditory-task Dataset**    In Section 3.4, we evaluate whether our model, trained on resting-state data, can generalize to a different domain without additional training. To this end, we use only the test set from the auditory-task EEG-fMRI dataset from Li et al. (2024b). During the scans, binaural tones were presented with randomized inter-stimulus intervals (ISI), and the task included two versions differing only in tone timing: (1) a fast-ISI version (500 TRs per scan) and (2) a sparse-ISI version (693 TRs per scan). The test set comprises four scans. Two scans correspond to the fast-ISI version, and the other two to the sparse-ISI version. For additional details, please refer to Li et al. (2024b).

**Vigilance States**    The vigilance state is represented as a categorical label with three classes—drowsy, intermediate, and alert—assigned to each fMRI frame. To derive these vigilance classes, we employed the automated VIGALL method (Huang et al., 2015; Jawinski et al., 2019; Olbrich et al., 2015), which classifies scalp EEG segments into five vigilance stages based on spatial power distributions. Specifically, we used the VIGALL 2.1 add-on in BrainVision Analyzer 2 to segment the preprocessed EEG into non-overlapping 1-second intervals and label each interval as one of five stages: A1, A2, A3, B1, or B2/3, corresponding to decreasing levels of alertness. Prior to staging, EEG signals were re-referenced to the average, and spherical spline interpolation was applied to reconstruct any missing channels required by the VIGALL standard. These vigilance stage labels are then grouped into 63-second epochs (corresponding to 30 fMRI time points) and the distribution of stages within each epoch is used to assign a single vigilance class, i.e., alert, intermediate, or drowsy, to that epoch, according to the following rules: (1) First, the five VIGALL stages were converted to integer values from 1 (most drowsy) to 5 (most alert); (2) The Wilcoxon signed-rank test was then applied to the integer values of each epoch to test for a significant difference of the median away from a (weighted) center value of 2.75; (3) Based on the test statistic, we assigned each epoch to one of the three vigilance classes using a z-threshold of ±1.5: epochs with significantly high or low median vigilance were labeled as alert or drowsy, respectively, while others were classified as intermediate; (4) Finally, consecutive epochs with the same vigilance label were merged to form continuous segments.

## C   MORE IMPLEMENTATION DETAILS

### C.1   HYPERPARAMETERS

The default hyperparameters for the full pretraining model architecture are summarized in Table 3. The fMRI masked signal modeling (f-MSM) model is pretrained for *225 K* iterations. The checkpoint achieving the highest spatial correlation between predicted and ground-truth signals on the validation set is selected as the final pretrained model, which is further fine-tuned for 20 epochs. The training hyperparameters of EEG-to-fMRI mapping are shown in Table 4.

**Other Implementation Details**    We initialize EEG encoder's spatiotemporal module using pre-trained weights from LaBraM-base (Jiang et al., 2024), with a token length of 200 (i.e., 1 second) and no overlap. For the multi-scale spectral module, we set the smallest scale size to $l_0 = 200$ (1 seconds without overlap), and use a multiscale level of 3. The functional connectivity metrics, i.e., Conn PCorr and Conn MSE, are calculated using the upper triangle of the correlation matrices, as they are symmetric.

Table 3: Hyperparameters used for stage 1: f-MSM pretraining and finetuning

| Hyperparameters | Values |
|---|---|
| batch size | pretrain: 96, fine-tune: 16 |
| learning rate | pretrain: 3e-4, fine-tune: 5.3e-5 |
| weight decay | 0.05 |
| Optimizer | AdamW |
| patch size | 1 |
| encoder embedded dim | 512 |
| mask ratio | 0.5 |
| mlp ratio | 2.0 |
| decoder embedded dim | 256 |
| encoder depth | 12 |
| encoder heads | 8 |
| decoder depth | 8 |
| decoder heads | 8 |

Table 4: Hyperparameters for stage 2: EEG-to-fMRI mapping

| Hyperparameters | Values |
|---|---|
| Batch size | 64 |
| Peak learning rate | 3e-4 |
| Minimal learning rate | 1e-6 |
| Learning rate scheduler | Cosine |
| Optimizer | AdamW |
| Adam $\beta$ | (0.9,0.99) |
| Weight decay | 0.05 |
| Drop path | 0.1 |
| Layer-wise learning rate decay | 0.65 |
| $\lambda_{\mathrm{MSE}}$ | 0.5 |
| $\lambda_{\mathrm{align}}$ | 0.8 |
| $\lambda_{\mathrm{recon}}$ | 0.2 |

## C.2 BASELINES

We compare our model against three publicly available and adaptable EEG-to-fMRI translation frameworks, all of which have been benchmarked in Li et al. (2024b). These are the only open-source methods compatible with the datasets and experimental setup in this study and Li et al. (2024b).

- **BEIRA** (Kovalev et al., 2022): BEIRA introduces a convolutional neural network (CNN)-based encoder-decoder architecture that translates EEG sequences into corresponding fMRI sequences in a sequence-to-sequence manner.

- **Li et al.** (Li et al., 2024a): This method extends BEIRA by incorporating an additional light-weight spectral representation learning module that leveraging sinusoidal activation function to better capture the frequency characteristics of EEG signals. It uses CNN-based downsampling and upsampling encoder-decoder blocks to perform the translation from EEG to fMRI during an eyes-open-eyes-closed task.

- **NeuroBOLT** (Li et al., 2024b): NeuroBOLT proposes a transformer-based multi-dimensional encoder for EEG-to-fMRI mapping in a seq-to-one format. It is a region-specific model, which means that models are trained separately for each region. It achieved state-of-the-art prediction performance in their resting-state dataset.

Among these baselines, the models by BEIRA and Li et al. were originally designed in a sequence-to-sequence format, where both the input and output are time series. To account for the hemodynamic delay of fMRI relative to EEG, the EEG sequence was temporally shifted by 6 seconds, i.e., the input EEG was delayed by 6 seconds to align with the corresponding fMRI response. For now

we do not include CATD (Yao et al., 2025) as a baseline, since CATD operates at the surface-map level restricted to the cortex, whereas our study focuses on ROI-based reconstruction covering the whole brain, including both cortical and subcortical regions, making the two settings not directly comparable. Moreover, since the implementation of CATD has not been publicly released, accurate reproduction is not currently feasible, which would preclude a fair comparison.

We also compare our model performance with state-of-the-art EEG encoders, and results are shown in D.1.

- **CBraMod** (Wang et al., 2025): CBraMod is a recent foundation model for EEG that follows the design of prior EEG foundation models by segmenting EEG signals into patches and pre-training via masked patch reconstruction. Building on this framework, CBraMod introduces two key innovations: (1) a criss-cross transformer backbone with parallel spatial and temporal attention mechanisms to separately capture heterogeneous dependencies in EEG, and (2) an asymmetric conditional positional encoding scheme that enables flexible adaptation to diverse EEG formats. Pre-trained on a large-scale EEG corpus, CBraMod outperforms state-of-the-art methods and demonstrates strong generalizability across up to 10 downstream BCI tasks (12 public datasets). In our experiments, we initialize the model with these pre-trained weights to provide a warm start for the EEG-to-fMRI translation task.

- **LaBraM** (Jiang et al., 2024): LaBraM (Large Brain Model) is a unified foundation model for EEG that enables cross-dataset learning by segmenting EEG signals into channel patches and using vector-quantized neural spectrum prediction to encode them into compact neural codes. Pre-trained on 2,500 hours of EEG data from 20 datasets, LaBraM achieves state-of-the-art performance in various downstream tasks such as abnormal detection, event classification, emotion recognition, and gait prediction. In our experiment, we load the pre-trained weights as initialization (version: LaBraM-base).

- **BIOT** (Yang et al., 2023): BIOT is a transformer-based foundational architecture for biomedical signal encoding. It segments EEG signals into patches and learns spatiotemporal and spectral representations from EEG, which can be applied to various downstream tasks.

- **CNNTransformer** (Peh et al., 2022): CNNTransformer is a transformer convolutional neural network originally designed for automated artifact detection in EEG.

- **STTransformer** (Song et al., 2021): STTransformer is a transformer-based spatial-temporal feature learning neural network originally designed for EEG decoding.

- **FFCL** (Li et al., 2022): FFCL is a model combining learned latent features from CNN and LSTM models for the purpose of motor imagery EEG classification.

In the NeuroBOLT experiments Li et al. (2024b), the authors adapted all baselines to a sequence-to-one format for evaluation. Following this approach, we apply the same adaptation and further attach a shared multi-ROI MLP decoder to each EEG encoder, enabling a single model to predict the full set of ROI signals for fair comparison. Despite using pretrained EEG foundation model encoders, all baseline models are retrained from scratch to convergence under the same training protocol as ours (including optimizer, batch size, number of epochs, learning-rate schedule, and early stopping criteria). All baselines reached stable convergence under this standardized setup.

# D  ADDITIONAL RESULTS

## D.1  PREDICTION PERFORMANCE OF OTHER EEG ENCODING BASELINES

In this section, we compare the performance of our model with that of other state-of-the-art EEG encoders, which were originally developed either as general-purpose EEG foundation models (Yang et al., 2023; Jiang et al., 2024; Wang et al., 2025) or were specifically designed for EEG decoding tasks (Song et al., 2021; Peh et al., 2022; Li et al., 2022). For a fair comparison, we attach a multi-region prediction head to each encoder to decode the full vector of fMRI ROIs. The results are reported in Table 5, where our model achieves superior performance on the majority of evaluation metrics.

Table 5: Full brain fMRI reconstruction: comparison with EEG encoding baselines. FB: Full brain; GM: Cortical gray matter; SC: Sub-cortical regions; CB: Cerebellum; Conn: Metric is applied on the upper triangle of the full-brain functional connectivity (FC) matrix; TCorr: Temporal correlation between predicted and ground truth fMRI signals; PCorr: Pixel-wise correlation between predicted and measured FC. **Bold**: the best; Underlined: the second best. Values are shown as mean ± std.

| Model Name | FB TCorr ↑ | GM TCorr ↑ | SC TCorr ↑ | CB TCorr ↑ | Conn PCorr ↑ | Conn MSE ↓ |
|---|---|---|---|---|---|---|
| Ours | **0.367 ± 0.052** | **0.394 ± 0.060** | **0.276 ± 0.082** | 0.247 ± 0.060 | **0.527 ± 0.084** | **0.233 ± 0.072** |
| CBraMod (Wang et al., 2025) | 0.349 ± 0.033 | 0.369 ± 0.035 | **0.276 ± 0.081** | **0.257 ± 0.059** | 0.520 ± 0.062 | 0.234 ± 0.072 |
| LaBraM-base (Jiang et al., 2024) | 0.288 ± 0.041 | 0.320 ± 0.049 | 0.207 ± 0.082 | 0.161 ± 0.037 | 0.432 ± 0.069 | 0.401 ± 0.090 |
| BIOT (Yang et al., 2023) | 0.318 ± 0.068 | 0.353 ± 0.075 | 0.211 ± 0.101 | 0.175 ± 0.053 | 0.489 ± 0.077 | 0.299 ± 0.089 |
| CNNTransformer (Peh et al., 2022) | 0.319 ± 0.085 | 0.346 ± 0.092 | 0.242 ± 0.083 | 0.197 ± 0.081 | 0.518 ± 0.078 | 0.281 ± 0.070 |
| STTransformer (Song et al., 2021) | 0.091 ± 0.062 | 0.106 ± 0.074 | 0.052 ± 0.036 | 0.048 ± 0.015 | 0.436 ± 0.075 | 0.326 ± 0.033 |
| FFCL (Li et al., 2022) | 0.298 ± 0.034 | 0.321 ± 0.030 | 0.220 ± 0.074 | 0.194 ± 0.048 | 0.471 ± 0.075 | 0.319 ± 0.085 |

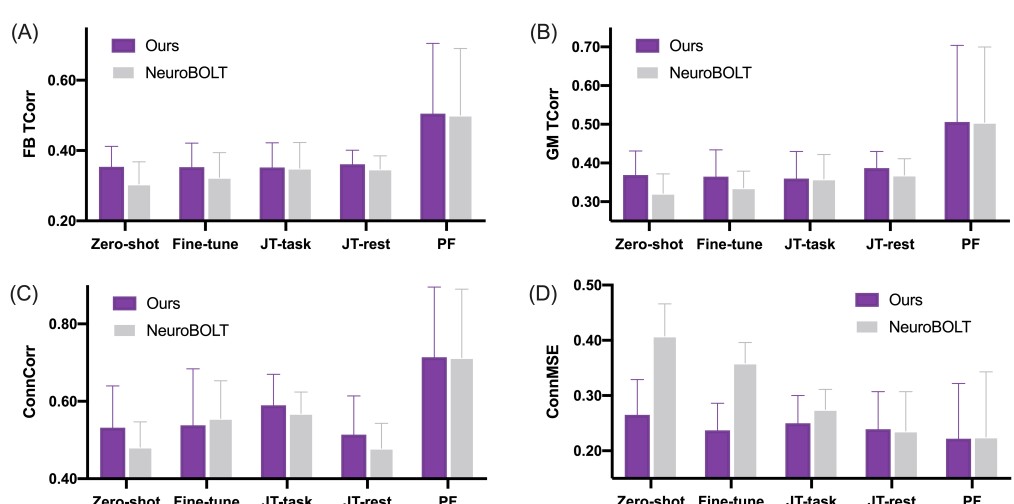

Figure 5: **Comparison across rest-to-task transfer and training strategies.** Performance comparison between our model and the state-of-the-art EEG-to-fMRI translation baseline, NeuroBOLT, under various evaluation setups: zero-shot transfer, fine-tuning, joint training on both rest and task scans with testing on task scans (JT-task), joint training and testing on resting-state scans (JT-rest), and personalized fine-tuning on individual task scans using a model pretrained on resting-state data (PF). (A) Full-brain temporal correlation (FB TCorr); (B) Gray matter temporal correlation (GM TCorr); (C) Spatial correlation of predicted and ground-truth functional connectivity (ConnCorr); (D) MSE of connectivity strength between real and reconstructed FC matrices (ConnMSE).

## D.2 RESTING-STATE TO TASK-CONDITION GENERALIZATION

In this section, we include a more detailed evaluation of generalization from resting-state EEG-fMRI to task-based EEG-fMRI using the auditory task dataset in Li et al. (2024b). We followed the same train-test split, resulting in 9 scans for training, 3 for validation, and 4 for testing. The results are summarized below in Figure 5 and benchmarked against the current state-of-the-art method, NeuroBOLT.

Specifically, we considered four experimental settings: **(1) Zero-shot generalization:** The model is trained only on resting-state data and directly evaluated on task fMRI without any further training. **(2) Fine-tuning:** The model is pretrained on resting-state data and then fine-tuned on task data. **(3) Joint training:** The model is trained on a mixture of resting-state and task data. **(4) Personalized fine-tuning:** Starting from the resting-state pretrained model, we fine-tune individually for each test scan in the task dataset using 80% of the scan for fine-tuning, 10% for validation, and 10% for testing. As shown in Figure 5, our model shows strong generalization from resting-state to task-based fMRI in the zero-shot setting. Fine-tuning improves performance slightly, mainly in the FC reconstruction part. Joint training helps task-fMRI FC reconstruction, but not necessarily for resting-state, which

might be due to already richer variability of brain dynamics in the resting state. Although both models perform similarly in personalized fine-tuning, overall our method still performs better in most metrics especially in full-scan reconstruction scenario.

### D.3 Connectome Fingerprinting Validation of Zero-Shot FC Reconstruction

Table 6: Connectome fingerprinting accuracy across brain regions using ground-truth and zero-shot predicted fMRI. Our model's zero-shot outputs preserve subject-specific FC signatures. FB: full-brain; GM: gray matter; SC: subcortical regions; FC: functional connectivity; Acc: fingerprinting accuracy.

| Model | FB-FC Acc | GM-FC Acc | SC-FC Acc |
|---|---|---|---|
| Ground-truth fMRI | $100\% \pm 0\%$ | $100.00\% \pm 0.00\%$ | $100\% \pm 0\%$ |
| Zero-shot pred fMRI | $90\% \pm 10\%$ | $80.00\% \pm 14.14\%$ | $90\% \pm 10\%$ |

To further validate the quality of the zero-shot generated fMRI under task conditions as described in D.2, we performed a connectome fingerprinting analysis (Finn et al., 2015). This approach assesses whether the predicted functional connectivity (FC) patterns retain one's true brain portrait and subject-specific patterns by attempting to identify individuals based on their FC profiles.

Specifically, we selected 5 subjects from the auditory task dataset, each of whom had two scans under **different conditions (fast and sparse auditory stimulus)**. In each trial, one scan per subject was randomly assigned to a "database set", and the other scan formed the "target set". For each target FC matrix, we computed Pearson correlations with all database matrices (using vectorized upper-triangular edge values), and predicted subject identity by selecting the database matrix with the highest similarity. This procedure was repeated across all 16 possible permutations (trials), and the average identification accuracy was reported.

As shown in Table 6, ground-truth fMRI achieved perfect identification accuracy. Notably, our model's zero-shot predicted fMRI also demonstrated high fingerprinting accuracy across full-brain, gray matter, and subcortical FC matrices. These findings suggest that the generated fMRI signals preserve individualized functional connectivity signatures, supporting their potential utility in downstream applications involving subject-specific brain representations, such as cognitive trait identification, behavioral decoding, and clinical profiling (Finn et al., 2015; Mantwill et al., 2022; Lu et al., 2024).

### D.4 Detailed Evaluation of Rest-to-task Zero-shot Transfer

We provide a comprehensive analysis of the zero-shot fMRI reconstruction performance in Figure 6. Specifically, panel (A) visualizes the region-wise temporal correlation across all 512 ROIs, and panel (B) summarizes these results (TCorr values) by averaging ROIs within each corresponding brain network to highlight network-level reconstruction quality.

In addition to these region-aggregated TCorr scores, we further evaluate network-level time-series reconstruction (see Figure 6(C-D)). Specifically, for each of the 17 networks defined in the Yeo atlas, we directly average the reconstructed ROI time series within that network to obtain a single network-level time series per subject. This provides a complementary assessment that focuses on larger-scale temporal dynamics rather than ROI-level variations. Finally, we include example reconstructions from the Salience/Ventral Attention Network, which contains several deep and non-surface regions (e.g., anterior insula, dorsal ACC). These examples illustrate that our model is capable of recovering coherent temporal structure even in networks that are typically challenging for EEG-based methods.

### D.5 Zero-shot Evaluation on Broader Tasks

To more thoroughly evaluate the model's zero-shot generalization ability beyond the auditory task included in the paper, we conducted two additional zero-shot analyses on publicly available datasets, targeting both paired EEG–fMRI settings and real-world EEG-only scenarios.

**Parkinson's disease EEG dataset** To evaluate real-world applicability where fMRI is unavailable, along with generalizability to a clinical population, we tested our model on a Parkinson's EEG

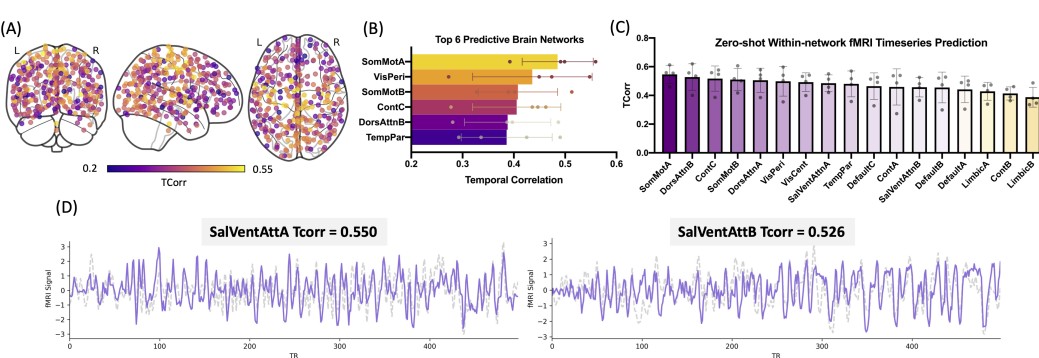

Figure 6: **Zero-shot fMRI reconstruction performance.** (A) Visualization of reconstruction performance across the whole brain; dots represent brain regions. (B) Top predictive brain networks; dots represent 4 evaluation participants in task dataset. (C) Network-wise fMRI timeseries reconstruction performance; dots represent 4 evaluation participants in task dataset. (D) Example visualization of zero-shot reconstructed network-wise fMRI time within Salience/Ventral Attention Network (gray: ground-truty, purple: prediction).

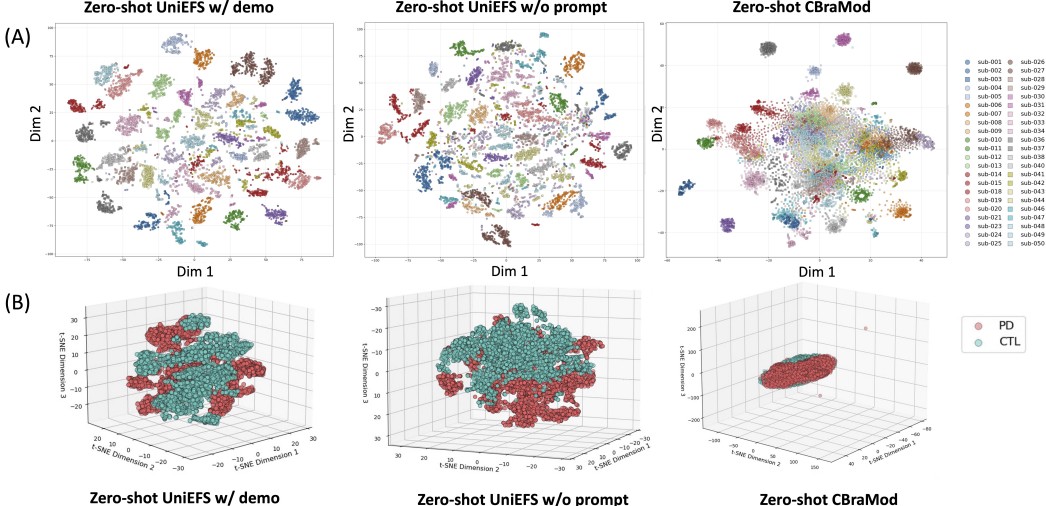

Figure 7: **Zero-shot EEG latent generated by UniEFS and CBraMod**. (A) 2-D t-SNE with each subject visualized with different colors. (B) 3-D t-SNE with colors distinguishing Parkinson patient (PD) and control (CTL) groups

dataset (Cavanagh, 2021) with task conditions (oddball task) and demographic distribution (elderly participants) entirely different from our training data. We refer the reader to the dataset paper for details of the EEG collection, task condition, and demographic information.

For this raw dataset, we performed standard EEG preprocessing, including band-pass filtering (0.1–75 Hz), 60 Hz notch filtering, and ICA-based artifact removal to eliminate eye-related components. The cleaned EEG was then epoched into 16-seconds windows with a 2-second stride (i.e., 14-second overlap), and fed each window into our trained UniEFS model to extract the fMRI-informed EEG latent embedding. We then compared the t-SNE plots of our model's embeddings versus the embeddings from the state-of-the-art EEG foundation model CBraMod (Wang et al., 2025) as shown in Figure 7. In summary, we have following key observations:

(1) **Clear separation between Parkinson patients and controls:** In the zero-shot setting, our latent representations cleanly separate the PD group from the control group, without any finetuning. This indicates that the learned EEG→fMRI projection extracts neurophysiologically meaningful structure that generalizes to entirely new populations. Such zero-shot group separation suggests promising potential for downstream clinical applications (e.g., biomarker discovery, screening, or disease monitoring) even when no fMRI is available.

(2) **Strong individual-specific clustering:** Samples from the same subject cluster tightly together in our latent space, indicating that the EEG→fMRI projection learns highly individualized EEG representations even without explicit subject labels. Incorporating demographic embeddings (age, sex) makes these subject-wise clusters even more compact, suggesting that the model can disentangle individual-specific neural signatures from population-level factors. This pattern is consistent with our fingerprinting results reported in the main paper, where subject identity can be reliably recovered from the predicted fMRI signals. From the latent-space perspective, this further demonstrates that the learned representation is not only discriminative but also structured in a way that preserves stable, subject-specific traits. This also highlights the potential value of our representation for various downstream applications.

(3) **Better zero-shot structure than CBraMod:** Compared to CBraMod, our embeddings show clearer intra-subject consistency and stronger inter-group separability under a completely zero-shot setting, even though our model is trained on only 28 scans rather than a large-scale EEG corpus. This highlights that the performance does not stem from data scale, but from the inductive bias introduced by learning an fMRI-augmented EEG embedding, where projecting EEG into the semantically structured fMRI latent space provides richer physiological constraints and induces robust, discriminative, and subject-specific EEG representations, even in the absence of paired fMRI during inference. This opens the possibility for using fMRI-informed EEG representations as a scalable backbone for future EEG-only downstream tasks.

**Paired EEG-fMRI dataset for motor conditions**   We evaluated our model pretrained using resting-state data on the Simultaneous EEG–fMRI Dataset for Multiple Motor Conditions (Bondi et al., 2025), which contains task paradigms completely different from our auditory setting. As demographic information is not provided for this dataset, we evaluated the pretrained model without prompt conditioning. The dataset paper contains full details on EEG–fMRI acquisition and preprocessing, which we refer the reader to for addtional information. Similarly, we directly fed the segmented EEG window into our model pretrained on resting-state and assess the reconstructed fMRI signals. As shown in Table 7, zero-shot predictions already produce meaningful temporal correlations and connectivity structure, demonstrating stable cross-dataset generalization even when the task paradigm and participant demographics differ substantially from training. Finetuning further improves subcortical and cerebellar reconstruction (SC TCorr: 0.158→0.233; CB TCorr: 0.195→0.213) and substantially strengthens connectivity estimation (ConnCorr: 0.363→0.509; ConnMSE: 0.257→0.139).

In summary, these analyses together provide additional support that our model generalizes to novel task structures, new subjects, and unseen EEG–fMRI dynamics.

### D.6    LEAVE-ONE-SITE-OUT TEST

Figure 8 presents a detailed comparison between leave-one-site-out zero-shot generalization and within-site training with three datasets (Rest Dataset 1, Rest Dataset 2, and Auditory-task Dataset), along with a comprehensive ablation of the different prompt components. Panel (A) shows that

Table 7: Zero-shot vs. finetuned performance across metrics for motor-task fMRI reconstruction.

| Condition | FB TCorr | GM TCorr | SC TCorr | CB TCorr | ConnCorr | ConnMSE |
|---|---|---|---|---|---|---|
| Zero-shot | $0.237 \pm 0.083$ | $0.266 \pm 0.087$ | $0.158 \pm 0.083$ | $0.195 \pm 0.091$ | $0.363 \pm 0.114$ | $0.257 \pm 0.024$ |
| Finetuned | $0.254 \pm 0.052$ | $0.272 \pm 0.057$ | $0.233 \pm 0.064$ | $0.213 \pm 0.059$ | $0.509 \pm 0.060$ | $0.139 \pm 0.013$ |

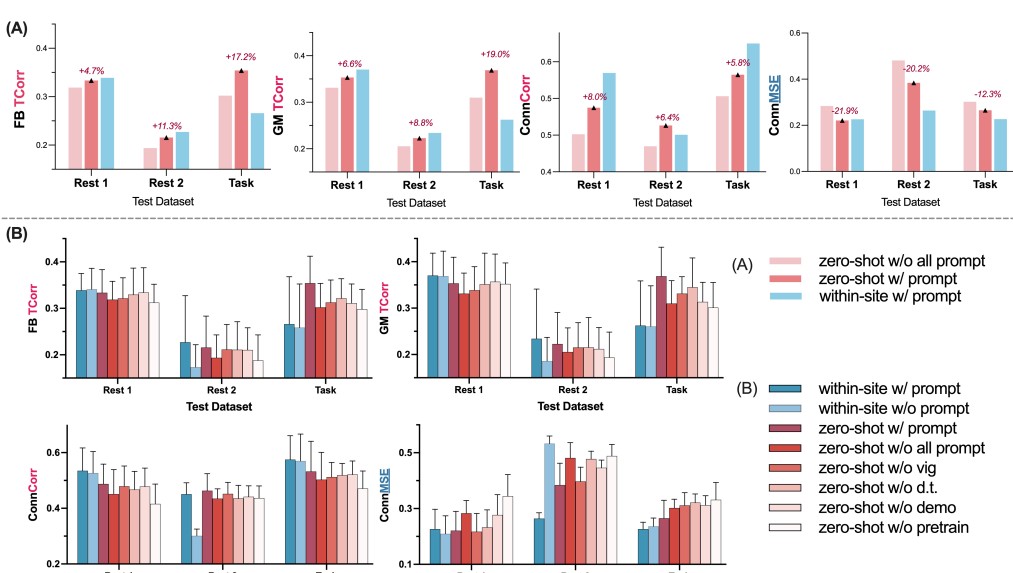

Figure 8: **Evaluation of leave-one-site-out vs. within-site performance under different prompt ablation settings.** (A) Zero-shot leave-one-site-out performance compared with within-site training for the full prompt model and the model without any prompts. Each bar shows the mean reconstruction metric across subjects, with percentage values indicating the improvement from adding prompts.(B) Comprehensive ablation analysis of each prompt component, including dataset tokens (d.t.), vigilance (vig), demographic prompts (demo), and pretraining, evaluated under both zero-shot and within-site settings. Error bars denote subject-wise standard deviation. Across test datasets (Rest 1, Rest 2, and Task), prompts consistently improve zero-shot generalization, and removing specific prompt types or pretraining reveals their individual contributions to mitigating cross-site domain shifts.

adding prompt tokens consistently improves zero-shot performance across all metrics and datasets, reducing the cross-site performance gap and in several cases even matching or surpassing within-site results. This indicates that the prompt embeddings effectively capture site-, demographic-, and vigilance-related variability, enabling robust generalization when testing on unseen sites.

Panel (B) further decomposes the contribution of each prompt type. Removing individual prompts (dataset tokens, vigilance, demographic cues) leads to consistent drops in zero-shot performance, highlighting that each component contributes complementary contextual information. Notably, removing the pretraining stage causes the substantial degradation, confirming that fMRI self-supervised pretraining is essential for stabilizing cross-site representations.

### D.7 PHYSIOLOGICAL PLAUSIBILITY CHECK

To further evaluate the physiological plausibility of the generated fMRI signal, we calculated and compared the power spectrum of real and generated fMRI signals across all regions. As shown in Figure 9, the predicted BOLD time series recovers the characteristic low-frequency PSD profile of real fMRI, including the dominant <0.1 Hz band and the expected log-linear decay. Moreover, the predicted and true PSDs exhibit very high correspondence across frequencies (Pearson r = 0.93, p< 0.001; Figure 9(B), demonstrating that the model captures the correct temporal dynamics underlying

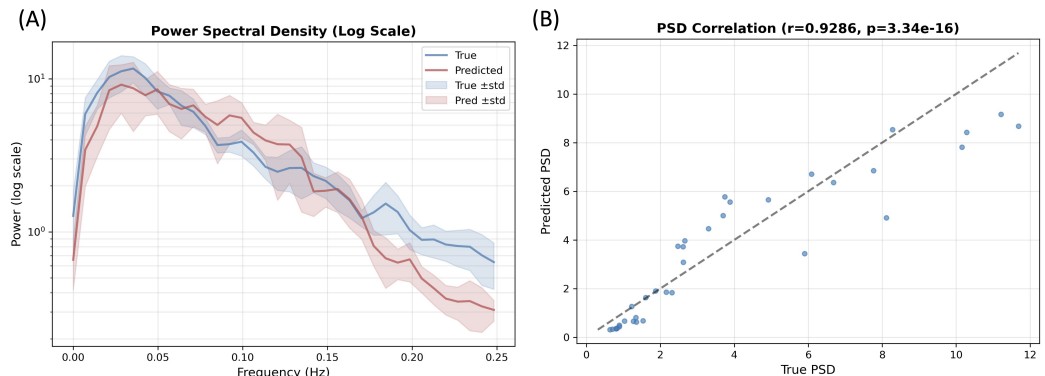

Figure 9: **Real and generated fMRI spectrum comparison**

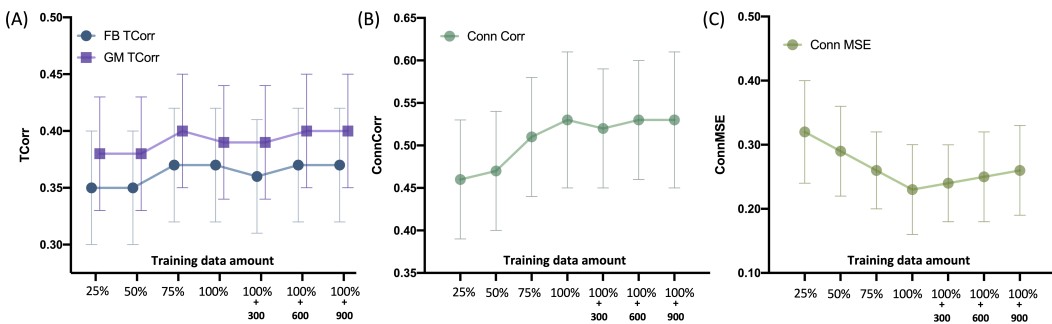

Figure 10: **Impact of data amount during pretraining** (under varying proportions of training scans: 25%, 50%, 75%, and 100%, where 100% corresponds to the dataset used in the main experiments; additional 300, 600, and 900 age-matched scans from HCP-A were also evaluated). (A) Impact on averaged temporal correlation within full-brain (FB TCorr) and gray matter (GM TCorr). (B) Spatial correlation (Conn Corr) between predicted and ground-truth functional connectivity (FC). (C) Mean squared error (Conn MSE) of connectivity strength in reconstructed FC matrices.

fMRI fluctuations. This analysis complements our TCorr and FC evaluations by confirming that the reconstructed signals preserve known neurophysiological properties of the BOLD response.

## D.8 EFFECT OF DATA SCALING DURING PRETRAINING

To further assess the impact of sample size during uni-modal fRMI pretraining, we conducted additional experiments by varying the number of training samples. We also incorporated age-matched resting-state scans (ages 35–50) from the HCP-Aging dataset (Bookheimer et al., 2019; Harms et al., 2018) as supplementary pretraining data, allowing the model to learn from a broader range of fMRI variability and provide a more comprehensive analysis. As shown in Figure 10, we observed that using less than 75% of the training data leads to a noticeable performance drop across all metrics. However, once the training set exceeded 75% of the total data, the improvement in average temporal correlation became marginal. Interestingly, functional connectivity reconstruction, especially ConnMSE, continued to improve more consistently with additional data within the HCP-YA dataset. In contrast, supplementing with age-matched HCP-Aging scans did not lead to further improvements in EEG-to-fMRI translation, suggesting that the current data scale (1,200 training scans, 600 TRs each) may already be sufficient for effective pretraining. Therefore, we conclude that the current number of subjects in the HCP-YA dataset provides a substantial and efficient scale for fMRI representation learning and for capturing instantaneous spatial dependencies across brain regions.

## D.9 EFFECT OF MASKING RATIO DURING PRETRAINING

Table 8 shows the impact of different mask ratios on performance. We found that a mask ratio of 0.5 yielded the best results. In typical Masked Autoencoder (MAE) training, a high mask ratio (e.g., 0.75) is often chosen to challenge the model to recover missing information and learn robust representations (He et al., 2022; Chen et al., 2023). However, in our case, a mask ratio of 0.75 did not yield the best performance for fMRI ROI data. The suboptimal performance of this high mask ratio may be due to the significant loss of inter-ROI correlation information, which is necessary for recovering full brain fMRI patterns. In other words, since we have already averaged the (voxel-wise) signals within regions to obtain the ROI data, much of the redundancy in voxel-wise signals has been reduced. Masking too much information hampers the model's ability to capture the relationships between ROIs, which are essential for meaningful fMRI-to-EEG mapping. Preserving spatial continuity and functional connectivity is critical for the model to learn accurate representations. While when the mask ratio is small, it makes the task too easy for the model to learn complex patterns and may overfit to the existing information, leading to suboptimal generalization and performance on unseen data.

Table 8: Influence of mask ratio in f-MSM

| Mask ratio | FB TCorr ↑ | GM TCorr ↑ | SC TCorr ↑ | CB TCorr ↑ | Conn PCorr ↑ | Conn MSE ↓ |
|---|---|---|---|---|---|---|
| 0.25 | $0.352 \pm 0.044$ | $0.378 \pm 0.048$ | $\underline{0.272 \pm 0.080}$ | $0.234 \pm 0.052$ | $0.511 \pm 0.078$ | $0.274 \pm 0.086$ |
| **0.50** | $\mathbf{0.367 \pm 0.052}$ | $\mathbf{0.394 \pm 0.060}$ | $\mathbf{0.276 \pm 0.082}$ | $\mathbf{0.247 \pm 0.060}$ | $\mathbf{0.527 \pm 0.084}$ | $\mathbf{0.233 \pm 0.072}$ |
| 0.75 | $\underline{0.357 \pm 0.058}$ | $0.385 \pm 0.066$ | $0.269 \pm 0.090$ | $0.241 \pm 0.061$ | $\underline{0.518 \pm 0.086}$ | $0.263 \pm 0.085$ |

## D.10 IMPACT OF PATCH SIZE

Table 9: Influence of patch size in f-MSM

| Patch size | FB TCorr ↑ | GM TCorr ↑ | SC TCorr ↑ | CB TCorr ↑ | Conn PCorr ↑ | Conn MSE ↓ |
|---|---|---|---|---|---|---|
| **1** | $\mathbf{0.367 \pm 0.052}$ | $\mathbf{0.394 \pm 0.060}$ | $\underline{0.276 \pm 0.082}$ | $\mathbf{0.247 \pm 0.060}$ | $\mathbf{0.527 \pm 0.084}$ | $\mathbf{0.233 \pm 0.072}$ |
| 2 | $0.347 \pm 0.069$ | $0.375 \pm 0.079$ | $0.256 \pm 0.088$ | $0.222 \pm 0.057$ | $0.475 \pm 0.083$ | $0.322 \pm 0.088$ |
| 4 | $0.355 \pm 0.040$ | $0.378 \pm 0.045$ | $\mathbf{0.281 \pm 0.078}$ | $\underline{0.245 \pm 0.054}$ | $0.491 \pm 0.093$ | $0.282 \pm 0.081$ |
| 8 | $0.362 \pm 0.062$ | $0.387 \pm 0.070$ | $\mathbf{0.281 \pm 0.087}$ | $\underline{0.245 \pm 0.063}$ | $\underline{0.500 \pm 0.082}$ | $0.257 \pm 0.072$ |

In our default setting, the model uses a patch size of 1, where each token corresponds to a single brain region. This approach is grounded in the understanding that, unlike images - where adjacent pixels often share semantic content due to spatial continuity Dosovitskiy et al. (2020) - the ordering of regions of interest (ROIs) in a brain data vector does not inherently reflect anatomical proximity or functional similarity. Consequently, neighboring entries in the ROI vector may correspond to brain areas that are neither anatomically adjacent nor functionally related. By representing each ROI as a separate token, the model avoids imposing artificial spatial assumptions and allows for the learning of functional relationships based on actual connectivity patterns rather than from an arbitrary ordering. Here we compare the performance across different patch sizes in transformer of f-MSM. For patch sizes larger than 1, the patched data are transformed into embeddings using a 1D convolutional layer with the stride equal to the patch size. As shown in Table 9, a patch size of 1 achieves the best performance among most of the metrics compared with the larger patch sizes.

## D.11 COMPARISON BETWEEN DIFFERENT ALIGNMENT LOSS

Table 10 compares several alignment objectives used during Stage 2. Overall, MSE provides the most stable and accurate temporal reconstruction, achieving the highest or near-highest performance on FB TCorr, GM TCorr, and CB TCorr. While alternative objectives such as InfoNCE and cosine similarity show competitive performance on certain connectivity metrics (e.g., ConnCorr, ConnMSE), they exhibit substantially larger variance across subjects in subcortical regions, leading to potentially less reliable overall behavior.

Table 10: Experiment on different alignment losses.

| Loss Type | FB TCorr ↑ | GM TCorr ↑ | SC TCorr ↑ | CB TCorr ↑ | ConnCorr ↑ | ConnMSE ↓ |
|---|---|---|---|---|---|---|
| MSE (Ours) | **0.367 ± 0.052** | **0.396 ± 0.058** | 0.251 ± 0.0371 | **0.247 ± 0.060** | 0.527 ± 0.084 | 0.233 ± 0.072 |
| InfoNCE | 0.346 ± 0.028 | 0.372 ± 0.017 | **0.270 ± 0.270** | 0.217 ± 0.074 | 0.537 ± 0.167 | **0.180 ± 0.001** |
| Contrastive | 0.329 ± 0.005 | 0.354 ± 0.007 | 0.256 ± 0.441 | 0.207 ± 0.018 | 0.516 ± 0.150 | 0.204 ± 0.002 |
| Cosine | 0.362 ± 0.054 | 0.391 ± 0.063 | 0.266 ± 0.074 | 0.232 ± 0.053 | **0.558 ± 0.091** | 0.212 ± 0.071 |
| MSE+Cosine | 0.352 ± 0.050 | 0.379 ± 0.076 | 0.256 ± 0.435 | 0.236 ± 0.039 | 0.511 ± 0.153 | 0.246 ± 0.070 |

## D.12 ABLATION STUDY ON EEG ENCODER MODULES

Table 11 summarizes the ablation results on the EEG encoder. These results demonstrate that the encoder architecture is not arbitrary: the temporal–spatial module and the multi-scale spectral module capture distinct and complementary aspects of EEG signals, and both are empirically necessary for accurate EEG→fMRI reconstruction. This is consistent with the original NeuroBOLT findings and provides direct justification for our architectural design.

Table 11: Ablation study on architectural components. Mean ± std. Colors denote significance vs. full model dervied by paired t-test: red ($p<0.001$), yellow ($p<0.01$), blue ($p<0.05$). Bold indicates best performance per column.

| Model Variant | FB Corr ↑ | GM TCorr ↑ | SC TCorr ↑ | CB TCorr ↑ | ConnCorr ↑ | ConnMSE ↓ |
|---|---|---|---|---|---|---|
| Ours (Full) | **0.367 ± 0.052** | **0.396 ± 0.058** | 0.251 ± 0.0371 | **0.247 ± 0.060** | 0.527 ± 0.084 | 0.233 ± 0.072 |
| Ours (no MSS) | 0.189 ± 0.076 | 0.206 ± 0.092 | 0.126 ± 0.042 | 0.139 ± 0.074 | 0.122 ± 0.032 | 0.407 ± 0.098 |
| Ours (no TS) | 0.350 ± 0.065 | 0.376 ± 0.075 | **0.262 ± 0.083** | 0.226 ± 0.055 | **0.530 ± 0.074** | **0.230 ± 0.080** |

## D.13 IMPACT OF EEG INPUT LENGTH

Table 12: Performance under different EEG input lengths.

| EEG Length | FB TCorr ↑ | GM TCorr ↑ | SC TCorr ↑ | CB TCorr ↑ | ConnCorr ↑ | ConnMSE ↓ |
|---|---|---|---|---|---|---|
| 16 s | **0.367 ± 0.052** | **0.394 ± 0.060** | **0.276 ± 0.082** | **0.247 ± 0.060** | 0.527 ± 0.084 | 0.233 ± 0.072 |
| 12 s | 0.344 ± 0.049 | 0.368 ± 0.057 | 0.264 ± 0.069 | 0.235 ± 0.054 | **0.548 ± 0.065** | **0.187 ± 0.058** |
| 8 s | 0.303 ± 0.059 | 0.330 ± 0.067 | 0.225 ± 0.055 | 0.211 ± 0.047 | 0.519 ± 0.074 | 0.246 ± 0.077 |
| 4 s | 0.157 ± 0.030 | 0.157 ± 0.044 | 0.180 ± 0.039 | 0.166 ± 0.043 | 0.314 ± 0.040 | 0.249 ± 0.057 |

## D.14 PERFORMANCE WITH DIFFERENT DIFUMO ATLAS GRANULARITIES

Table 13 reports EEG-to-fMRI reconstruction performance across different DiFuMo atlas granularities (256, 512, and 1024 ROIs). Overall, the 256-ROI atlas yields the highest reconstruction accuracy across all metrics, followed by 512 ROIs, while 1024 ROIs shows the largest performance drop. This trend reflects the increasing difficulty of predicting finer-grained, higher-dimensional fMRI representations from limited EEG information: as ROI granularity increases, each region becomes smaller and noisier, making both temporal and connectivity reconstruction more challenging. These results suggest that moderate atlas resolutions (e.g., 256-512) strike a favorable balance between spatial detail and predictable signal quality.

## D.15 UNSEEN RESTING-STATE SCAN RECONSTRUCTION EXAMPLES

In this section, we present several examples of held-out resting-state fMRI scan reconstructions, focusing on regions similar to those displayed in Li et al. (2024b) (Figure 11). These examples demonstrate the ability of our unified model to efficiently reconstruct entire resting-state fMRI scans across a wide time range using a single model.

Table 13: Performance with different DiFuMo atlas granularities.

| ROI | FB TCorr ↑ | GM TCorr ↑ | SC TCorr ↑ | CB TCorr ↑ | ConnCorr ↑ | ConnMSE ↓ |
|---|---|---|---|---|---|---|
| 256 | **0.387 ± 0.059** | **0.415 ± 0.064** | **0.325 ± 0.084** | **0.260 ± 0.074** | **0.591 ± 0.076** | **0.213 ± 0.088** |
| 512 | 0.367 ± 0.052 | 0.396 ± 0.058 | 0.251 ± 0.037 | 0.247 ± 0.060 | 0.527 ± 0.084 | 0.233 ± 0.072 |
| 1024 | 0.321 ± 0.036 | 0.358 ± 0.043 | 0.262 ± 0.072 | 0.204 ± 0.058 | 0.485 ± 0.059 | 0.304 ± 0.072 |

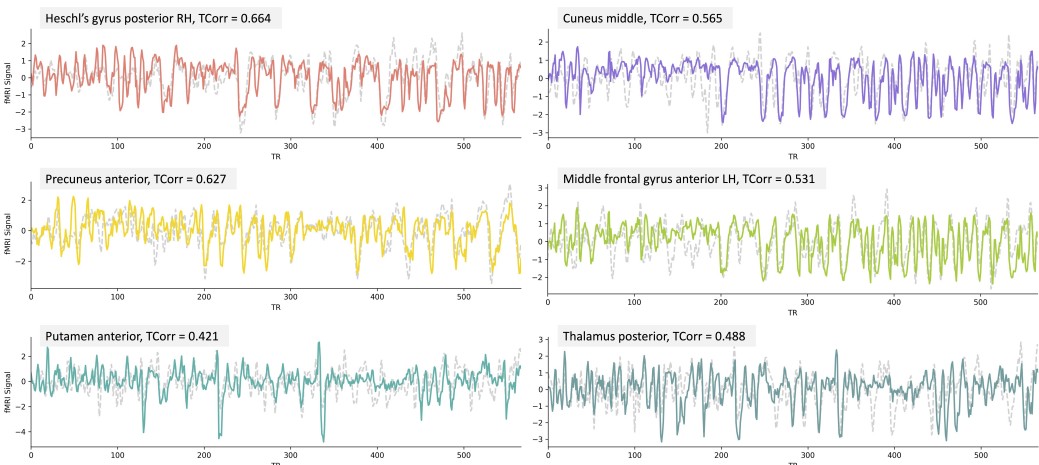

Figure 11: Held-out whole-scan reconstruction examples.

# E ADDITIONAL DISCUSSION

## E.1 DISCUSSION ON FC RECOVERY AND COMPARISON WITH BASELINES

While comparing our model with other EEG-to-fMRI translation baselines, we observe that UniEFS achieves the second-best performance in recovering functional connectivity (FC). One plausible reason our model does not outperform the CNN-based approach by Li et al. (2024a) in FC reconstruction is that FC is computed using Pearson correlation, which is highly sensitive to noise. Even minor prediction deviations can result in amplified discrepancies in pairwise correlations. The CNN baseline tends to produce smoother and more regularized outputs, which may suppress high-frequency fluctuations and thus yield more stable FC metrics—particularly in small-scale evaluation settings. In contrast, our model prioritizes frame-wise fidelity and regional dynamics, which may introduce local variability despite capturing more detailed temporal patterns. Notably, despite this, our model offers overall more consistent and strong performance across diverse evaluation settings.

## E.2 DISCUSSION ON THE COMPLEMENTARY ROLES OF RECONSTRUCTION AND ALIGNMENT LOSS

From Table 2, we observe that during the alignment stage the reconstruction loss contributes the largest performance gain, while the improvement from the alignment loss is numerically smaller. Importantly, however, the gains from the alignment term are statistically significant and consistently positive across all subjects, indicating that its effect is systematic rather than noise-driven. This behavior aligns with the distinct roles of the two losses in our architecture. In stage-2, the fMRI decoder is frozen; thus, the reconstruction loss provides the primary source of task-specific semantic supervision, guiding the EEG encoder to produce latent representations that remain decodable through the fixed decoder. By contrast, the alignment loss encourages geometric proximity between EEG and fMRI latents but does not account for the decoder's nonlinear inversion geometry. Because the pretrained fMRI latent space forms a curved and structured manifold, even small off-manifold deviations - while close in Euclidean distance - may decode into semantically incorrect fMRI patterns. Consequently, the reconstruction loss naturally leads to larger numerical improvements, whereas the alignment loss serves as a regularizer that improves latent-space geometry, stability, and cross-subject

consistency. This explains why its contribution is more modest in magnitude yet remains statistically reliable.

### E.3 CLARIFICATION ON RECONSTRUCTION VS. FUTURE-FRAME PREDICTION

The model reconstructs the fMRI signal associated with the neural activity expressed in the preceding EEG window. This choice follows the physiology of the hemodynamic response, where the BOLD signal at time t primarily reflects neural events that occurred several seconds earlier. Accordingly, our formulation focuses on recovering the temporally aligned BOLD representation rather than predicting future fMRI states beyond what is supported by the EEG window. This differs from multi-step forecasting approaches, which aim to predict future frames and require additional considerations such as modeling fMRI autocorrelation and controlling temporal information leakage. Such forecasting extensions are conceptually distinct and would be an interesting future direction to explore.

### E.4 LIMITATIONS AND FUTURE WORK

For the resting condition, our model is trained on only two paired EEG-fMRI datasets with fewer than 32 EEG electrodes. The limited electrode coverage may impede the ability to accurately reconstruct signals from subcortical regions. Since our context-aware embeddings are designed to accommodate variability across datasets and populations, we plan to incorporate and collect additional resting-state datasets, ideally with denser EEG electrode coverage, to further enhance the model's capacity for capturing fine-grained fMRI spatial dynamics, particularly in deep brain structures. Moreover, based on our finding that the model exhibits strong zero-shot transfer ability, an exciting future direction is to evaluate the pretrained resting-state model on diverse task-based datasets. Such downstream evaluations would allow us to probe how well the learned EEG-to-fMRI mapping generalizes beyond resting conditions, potentially enabling task-specific decoding and offering new insights into the neural mechanisms that link spontaneous and task-evoked brain activity. In the longer term, this line of work may also open avenues for clinical applications, such as noninvasive brain decoding and monitoring of cognitive or pathological states in settings where fMRI is impractical.

## F THE USE OF LARGE LANGUAGE MODELS (LLMS)

We disclose that large language models (LLMs) were employed during manuscript preparation purely for non-substantive editing tasks (e.g. grammar polishing and phrasing refinements). All scientific content, analysis, experimental design, and writing decisions were conceived and carried out solely by the authors.

