# OpenReview forum: "Mind the State: Towards Unified, Context-Aware EEG-to-fMRI Synthesis"
_ICLR.cc/2026/Conference — Submitted to ICLR 2026_

### Official Review · Reviewer_h7XH · 2025-10-29

**Soundness:** 3
**Presentation:** 3
**Contribution:** 3
**Rating:** 6
**Confidence:** 4

**Summary:**

This paper introduces UniEFS (Unified EEG-to-fMRI Synthesis), a framework for translating EEG signals into high-resolution, temporally synchronized fMRI representations. The model operates in two stages: first, an fMRI pretraining phase using masked signal modeling (f-MSM) to learn spatial-temporal structure from large unlabeled fMRI data; second, a context-aware EEG-to-fMRI mapping that aligns EEG embeddings with the pretrained fMRI latent space using transformer architectures. UniEFS incorporates context prompts encoding vigilance, demographics, and dataset information to handle variability across subjects and acquisition settings. Extensive experiments demonstrate that the model outperforms prior EEG-to-fMRI baselines in temporal and functional connectivity correlations, generalizes zero-shot to unseen task fMRI datasets, and preserves individualized connectome patterns. Overall, the paper contributes a scalable, unified, and context-adaptive model for multimodal brain activity translation, advancing cross-modal neuroimaging representation learning.

**Strengths:**

1. Unified, frame-wise, whole-brain EEG→fMRI synthesis with context-aware prefix prompts (dataset, age, sex, vigilance) is a fresh and relevant formulation for heterogeneous, eyes-closed resting-state data. The two-stage route—masked fMRI pretraining + domain adaptation, then latent alignment—addresses data scarcity and improves spatial priors.

2. Solid baselines and component ablations show: (i) pretraining matters, (ii) alignment and reconstruction losses are both needed, (iii) vigilance/demographic/dataset tokens help, with quantifiable drops when removed. Zero-shot task transfer and fingerprinting analyses further speak to generalization.

3. The method is structured and motivated (why ROI-level, why high mask ratio, why prompts), with clear training pipeline and evaluation. Figures highlighting network-wise effects of vigilance conditioning are compelling and biologically plausible.

4. Demonstrates consistent improvements over prior EEG→fMRI baselines in temporal correlation across cortical/subcortical/cerebellar regions, with potential downstream impact for scalable, portable neuroimaging in clinics and naturalistic settings

**Weaknesses:**

1. The paired EEG-fMRI datasets are relatively small (22+7 subjects, 29+10 scans), which may limit claims of broad generalization; additional public cohorts or cross-site test-only evaluations would strengthen external validity.

2.  While prompts encode context, the paper lacks deeper analyses of site/scanner/TR shifts and how much prompts vs. pretraining mitigate them (per-site performance, leave-one-site-out).

3. Leakage/overfitting risks: The domain adaptation step fine-tunes on the fMRI portion of the paired set; more guardrails (strict subject-held-out splits are used, but clarifying leakage risks and testing strict cross-dataset generalization would help).

4. Emphasis is on TCorr and FC correlations; adding voxel/surface recon quality (where applicable), physiological plausibility checks, and per-network temporal dynamics metrics (e.g., dynamic FC) in main text would round out evaluation.

5. Statistical rigor in main text: Some results rely on means±sd; more confidence intervals, multiple-comparison controls, and explicit p-values (beyond the single paired t-test note) in the main body would improve soundness.

**Questions:**

1. Can you report per-dataset/per-site results and leave-one-dataset-out evaluations to quantify true OOD generalization and the specific gains attributable to dataset tokens?

2. Beyond on/off, can you provide effect sizes for each prompt type across networks and subjects (e.g., vigilance vs. demographics), and whether prompts ever harm performance on homogeneous subsets?

3. In Table 2, removing Lrecon devastates performance. Could you include a matched variant using only Lrecon (no Lalign) and quantify stability/learning speed to better justify the latent alignment pathway?

4. Please expand zero-shot results with region/network-wise breakdowns and compare against personalized fine-tuning and adapter/LoRA to test whether EEG encoders or the fMRI decoder bottleneck limits transfer.

5. Did you try HRF-aware temporal shifts/warps or frequency-domain perturbations during Stage-2 training? How sensitive is performance to EEG window length (16s) and DiFuMo granularity (P=512)?

6. Any preliminary analyses that relate reconstructed fMRI to behavioral/clinical covariates (even post-hoc), to support applied value claims?

---

> ### Author Response · Authors · 2025-11-27
> **Author Rebuttal - Part 1**
>
> We sincerely thank the reviewer for the insightful review and encouraging feedback. Your constructive suggestions have greatly helped us improve the quality of the manuscript. We have carefully revised the manuscript according to your comprehensive suggestion and uploaded **a new version of the PDF**, with all updated or newly added content highlighted in ***blue***.
>
> Please find our point-by-point responses to your comments below:
>
> >**1. Further zero-shot evaluations on additional public cross-site cohorts and potential future clinical application (W1, Q6)**
>
> We thank the reviewer for raising this important point. We fully agree that larger cross-site cohorts would further strengthen external validity. At the same time, we would like to emphasize that the current dataset scale reflects the state of the field. Publicly available *simultaneous EEG-fMRI* datasets are extremely limited, and most existing cohorts contain fewer than 20 subjects, reflecting a common limitation across the literature rather than a constraint specific to our work. Gathering EEG data during fMRI studies requires specialized hardware (e.g., EEG amplifier that operates within a strong magnetic field), expertise in handling the specific MRI-induced artifacts in the EEG, and substantial additional setup time, and therefore is not common in neuroscience or clinical research. While posing a limitation with regard to model training, these practical challenges in concurrent EEG-fMRI acquisition further motivate studies that can infer fMRI-like information from EEG alone, in line with the goals of the present study.
>
> To more thoroughly evaluate the model’s zero-shot generalization ability beyond the auditory task included in the paper, we conducted **two additional zero-shot analyses on publicly available datasets**, targeting *both* paired EEG-fMRI settings and real-world EEG-only scenarios, where fMRI is not available. To avoid redundancy, we kindly refer the reviewer to our detailed zero-shot analyses summary provided in Response to ***Reviewer hvue***, “***4. Further zero-shot evaluations on broader task  (W4)***”, as well as the full new results and figures in the revised manuscript ***\<Appendix Section D.5. and Figure 7\>***.
>
> >**2. Comparison of zero-shot leave-one-site-out generalization versus within-site performance (W2, Q1)**
>
> We appreciate the reviewer’s constructive suggestion. In response, we carried out the requested evaluation comparing zero-shot leave-one-site-out generalization with standard within-site performance. Because the full table is quite large, we consolidated the results into a single figure for clarity. Please see **Figure 8** in ***\<Appendix Section D.6\>*** of our updated manuscript.
>
> The key insights are summarized as follows:
>
> 1) **Prompt tokens consistently improve cross-site generalization:** As shown in Fig.8 (A), adding prompt tokens produces **uniform improvements** in the leave-one-site-out setting across all metrics. This indicates that the contextual information encoded in the prompts helps reduce site/scanner/TR variability and stabilizes the mapping across acquisition domains.
> 2) **Different prompt types contribute differently, but all reduce domain shift:**
>     Fig.8 (B) provides per-metric breakdowns. Removing each prompt type produces a measurable drop in zero-shot performance, showing that vigilance- and demographic-related context each contribute non-redundant information
> 3) **Pretraining plays a major role in mitigating cross-site variability**. From Fig. 8 (B), we also observe that removing pretraining leads to a substantial drop in cross-site performance.
> 4) **Zero-shot rest-\>task transfer exceeds even within-site performance**. As shown in Fig. 8 (A), the reconstructed task-condition time series (FB TCorr, GM TCorr) achieve even higher accuracy in the zero-shot setting than in the within-site baseline. This can occur because multi-site resting-state training provides substantially broader variability than single-site task data: resting-state EEG-fMRI contains diverse spontaneous dynamics across frequencies and networks, exposing the model to a richer range of neural patterns than the more structured task condition. In contrast, transferring task-\>rest is more difficult because task-evoked data capture a narrower set of patterns and thus generalize less effectively to the more heterogeneous resting-state.

---

> ### Author Response · Authors · 2025-11-27
> **Author Rebuttal - Part 2**
>
> >**3. Clarification on subject-held-out split in finetuning and alignment (W3).**
>
> We appreciate the reviewer’s comment. We would like to clarify that there is no data leakage in any stage of our framework.
>
> - **Decoder fine-tuning uses only the training subjects.**
> During the domain adaptation stage, we use only the fMRI scans from the **training split**, with **train/validation/test subjects in the paired dataset strictly disjoint**. The model is fine-tuned for 20 epochs, and the checkpoint is selected **exclusively based on validation loss**, without accessing any fMRI data from test subjects at any point. We explicitly added this clarification to the manuscript to avoid potential ambiguity. The purpose of this fine-tuning step is solely to mitigate site- or device-related domain shifts between the pretraining dataset and the target dataset, and does not involve any information leakage from evaluation subjects.
> - **Domain adaptation provides only marginal improvements, confirming the absence of leakage.**
> As shown in Table 2 of the original manuscript, removing fine-tuning yields almost identical performance across metrics. Fine-tuning led to only a small improvement in the connectivity reconstruction metric (ConnMSE, from 0.27 to 0.23; lower is better), while temporal-correlation-based metrics (Tcorr, ConnCorr) remain stable. This further indicates that fine-tuning is not exploiting any leakage but simply helps adjust for domain differences between datasets.
>
> >**4. Physiological plausibility check and more analysis (W4).**
>
> Thank you for the thoughtful suggestions. We appreciate the reviewer’s perspective and agree that these evaluations are valuable complementary future directions. Below, we clarify which aspects can be meaningfully evaluated within our framework and which would require extensions beyond the scope of the current study.
>
> * Voxel/surface-level reconstruction is not directly supported by the model design. Our model reconstructs ROI-level fMRI time series (512 ROIs) rather than voxel/surface maps, as they offer improved SNR compared to voxel-level signals and support efficient, interpretable modeling.
>
> * To strengthen the physiological plausibility analysis, we have added a new spectral evaluation of the reconstructed fMRI signals (see **Fig. 9** in ***\<Appendix Section D.7\>* of the revised manuscript**). As shown in **Fig. 9(A)**, the predicted BOLD time series recovers the characteristic low-frequency PSD profile of real fMRI, including the dominant \<0.1 Hz band and the expected log-linear decay. Moreover, the predicted and true PSDs exhibit very high correspondence across frequencies (Pearson *r* \= 0.93, *p* \< 0.001; Fig. 9(B)), demonstrating that the model captures the correct temporal dynamics underlying fMRI fluctuations. This analysis complements our TCorr and FC evaluations by confirming that the reconstructed signals preserve known neurophysiological properties of the BOLD response.
>
> * Dynamic FC could in principle be derived from the reconstructed ROI time series. However, dFC is a higher-order measure whose stability depends heavily on windowing choices, temporal length, and the specific estimation procedure. A careful and extensive evaluation of dFC reliability would therefore require dedicated analyses beyond the scope of the current study.  Since our work focuses on accurate reconstruction of the underlying fMRI time courses themselves, i.e., the foundation upon which higher-order metrics like dFC are computed, we view dFC analysis as a meaningful but separate extension beyond the scope of the present study. We agree that this is an interesting future direction and appreciate the reviewer for highlighting it.
>
> **Instead**, we now additionally include a **network-wise temporal dynamics analysis** in the revised manuscript. Specifically, for each canonical functional network in Yeo's 17 network atlas, we average the ROI time series within the network to obtain a network-level signal and compute the temporal correlation between predicted and ground-truth network activity. We have also added this in our revised manuscript, along with region-wise and network-wise evaluations. Please see these new analysis in ***\<Appendix Section D.4., Figure 6\>***.

---

> ### Author Response · Authors · 2025-11-27
> **Author Rebuttal - Part 3**
>
> >**5. Statistical Reporting (W5)**
>
> Thank you very much for the suggestion. In the revised manuscript, for clarity and readability, we report paired t-test p-values for the key ablation comparisons and the pairwise comparisons with baselines. The significance levels are grouped into three categories (p \< 0.05, p \< 0.01, p \< 0.001) and visualized using a three-color scheme (green, yellow, red) in the tables. We chose not to apply multiple-comparison corrections such as FDR for the following reasons:
> - The evaluation cohort is small (6 subjects), and standard correction procedures can become overly conservative and unstable under such conditions;
> - The purpose of these tests is to assess whether each component yields consistent improvements across subjects, rather than to make strong population-level inferential claims about effect sizes.
>
> For transparency, we have added a note in the revised manuscript clarifying this statistical rationale. We also note that this practice is in line with all prior EEG-to-fMRI baselines and fMRI synthesis work [1-4].
>
> >**6. Question 2**
>
> Thank you for the thoughtful suggestion. We would like to refer the reviewer to the expanded cross-site and within-site prompt-ablation analyses provided in the newly added bar-graph results ***\<Fig. 8 in the Appendix Section D.6 of the revised manuscript\>***, which already compare each prompt component across multiple sites, conditions, and model variants. These analyses more directly capture the robustness of each prompt type under cross-site distribution shifts, which was the primary motivation for introducing contextual prompts.
>
> Regarding the question of whether prompts may harm performance on homogeneous subsets, we agree this is an interesting direction. However, identifying homogeneous sub-groups (e.g., age bands, tightly matched vigilance levels, or scanner-specific cohorts) in a statistically meaningful way would require substantially larger and more diverse datasets than what is currently available in simultaneous EEG-fMRI research. With the current cohort sizes, defining such sub-groups would not be stable or reliable. We see this as a valuable future extension once larger datasets become available, and we appreciate the reviewer for highlighting this opportunity.
>
> >**7. Question 3: Comparisons with No Alignment Loss**
>
> We thank the reviewer for the question. In our original manuscript, we have already included the comparison with using only Lrecon (removing Lalign) (please see **Table 2 in the manuscript**). In the revision, we further performed paired t-tests to confirm that the improvements obtained by adding Lalign are statistically reliable.
>
> | Model Type | FB Tcorr | GM Tcorr | Conn PCorr | Conn MSE |
> | :---- | :---- | :---- | :---- | :---- |
> | Full | **0.367 ± 0.052\*\*** | **0.394 ± 0.060\*** | **0.527 ± 0.084\*\*** | **0.233 ± 0.072\*\*** |
> | w/o L\_align | 0.339 ± 0.052 | 0.367 ± 0.055 | 0.502 ± 0.082 | 0.280 ± 0.083 |
>
> Significance：p\<0.05 (\**), p\<0.01 (*\*\*), p\<0.001(\*\*\*), uncorrected.
>
> From the above results, we would like to emphasize that although the numerical gain from the alignment loss is smaller, it is **statistically significant and consistently positive across all subjects**. This indicates that the alignment term provides reliable, systematic benefits rather than noise-level variation.
>
> This pattern is consistent with the roles of the two losses in our architecture. In Stage 2, the fMRI decoder is frozen; therefore, the reconstruction loss serves as the primary source of task-specific semantic supervision, guiding the EEG encoder to produce latent representations that remain decodable through the fixed decoder. In contrast, the alignment loss enforces geometric proximity between EEG and fMRI latents but does not account for the decoder’s nonlinear inversion geometry. Because the latent space learned in Stage 1 forms a curved nonlinear manifold, even small off-manifold deviations, though close in Euclidean distance, may decode into semantically incorrect fMRI signals.
>
> As a result, the reconstruction loss naturally yields larger numerical gains, while the alignment loss functions as a regularizer that improves latent-space geometry, stability, and consistency, explaining why its improvements are modest in magnitude yet statistically reliable. We appreciate the reviewer pointing this out and have now included this discussion in **\<Appendix Section E2\>** in the revised manuscript.

---

> ### Author Response · Authors · 2025-11-27
> **Author Rebuttal - Part 4**
>
> >**8. Question 4**
>
> Thank you for your question\! We break this down into two sub-questions:
> **(1) region/network-wise zero-shot breakdown**, and
> **(2) testing whether the fMRI decoder constitutes a bottleneck, via personalized decoder fine-tuning.**
>
> 1) **Region- and Network-Wise Zero-Shot Performance**
>    In the revised manuscript, we now provide a full breakdown of zero-shot performance across all 512 ROIs and Yeo-17 networks ***\<Appendix Section D.4, Figure 6\>***, and also the visualizations of the gains provided by the vigilance token ***\<Main text, Section 3.3, Figure.3\>***
> 2) **Does the fMRI decoder create a bottleneck?**
>    To directly test whether the pretrained fMRI decoder limits transfer, we performed a controlled experiment for personalized finetuning comparing **(i)** *Decoder Frozen (our default zero-shot setting)* vs.**(ii)** *Decoder Personalized Fine-tuned on Each Test Subject*. We fully finetune the fMRI decoder instead of using adapter/LoRA to fully explore the upper bound on the effect of decoder adaptation and fully isolate whether the decoder limits zero-shot transfer. The results are summarized in the table below. Specifically, we found across all metrics, fine-tuning yields negligible or negative change. Since decoder fine-tuning does not help, this indicates that: **1\)** the pretrained fMRI decoder might not be the bottleneck; **2\)** transfer limitations are more likely attributable to the EEG encoder, the modality gap, or cross-subject EEG variability rather than decoder capacity.
>
>
> | Condition | FB TCorr | GM TCorr  | ConnCorr  | ConnMSE |
> | :---- | :---- | :---- | :---- | :---- |
> | Decoder Frozen | 0.501 ± 0.200 | 0.506 ± 0.198 | 0.717 ± 0.185 | 0.222 ± 0.100 |
> | Decoder Trainable | 0.494 ± 0.203 | 0.491 ± 0.207 | 0.660 ± 0.226 | 0.228 ± 0.146 |
> | Mean Difference (Train-Frozen) | \-0.007 ± 0.018 | \-0.015 ± 0.019 | \-0.057 ± 0.048 | 0.006 ± 0.059 |
>
> If we have misunderstood your question in any way, we would appreciate any additional clarification.

---

> ### Author Response · Authors · 2025-11-27
> **Author Rebuttal - Part 5**
>
> >**9. Question 5**
>
> We thank the reviewer for the insightful questions. Our model is designed as a sequence-to-one architecture, where each fMRI TR is predicted from a 16-second EEG window before each fMRI frame collection. This design implicitly accounts for HRF delay and dispersion: the temporal window fully covers the canonical 4-6 s HRF lag, and the transformer’s attention mechanism learns data-driven temporal alignment within this window. Thus, the model does not rely on a single EEG time point but instead integrates information across the entire HRF-relevant temporal context. For this reason, explicit HRF-aware temporal shifting/warping is not necessary in our framework, as the temporal mapping is learned end-to-end by the model.
>
> In accordance with your suggestion, we implemented frequency-domain perturbations during stage-2 training, and evaluated their effects on model performance. We found that frequency perturbations lead to slightly lower temporal-correlation metrics across full-brain and regional evaluations, while producing improvements in connectivity-based metrics (ConnCorr and ConnMSE). This indicates that this is an very interesting and promising approach to induice noise during training for learning more robust represenataion especially in recovering the brain connectivity, which we will consider to use in our future exploration.
>
> | Model | FB TCorr | GM TCorr | SC TCorr | CB TCorr | ConnCorr | ConnMSE |
> | :---- | :---- | :---- | :---- | :---- | :---- | :---- |
> | Original | **0.367 ± 0.052** | **0.394 ± 0.060** | **0.276 ± 0.082** | **0.247 ± 0.060** | 0.527 ± 0.084 | 0.233 ± 0.072 |
> | With freq perturbations | 0.360 ± 0.046 | 0.391 ± 0.055 | 0.253 ± 0.066 | 0.233 ± 0.049 | **0.538 ± 0.060** | **0.200 ± 0.057** |
>
> - The summarized results comparing the performance under different EEG window lengths are provided below, where wee see the current length provide the best performance in TCorr, and the second best in the FC related metrics.
>
> | EEG length | FB TCorr | GM TCorr | SC TCorr | CB TCorr | ConnCorr | ConnMSE |
> | :---- | :---- | :---- | :---- | :---- | :---- | :---- |
> | 16s | **0.367 ± 0.052** | **0.394 ± 0.060** | **0.276 ± 0.082** | **0.247 ± 0.060** | 0.527 ± 0.084 | 0.233 ± 0.072 |
> | 12s | 0.344 ± 0.049 | 0.368 ± 0.057 | 0.264 ± 0.069 | 0.235 ± 0.054 | **0.548 ± 0.065** | **0.187 ± 0.058** |
> | 8s | 0.303 ± 0.059 | 0.330 ± 0.067 | 0.225 ± 0.055 | 0.211 ± 0.047 | 0.519 ± 0.074 | 0.246 ± 0.077 |
> | 4s | 0.157 ± 0.030 | 0.157 ± 0.044 | 0.180 ± 0.039 | 0.166 ± 0.043 | 0.314 ± 0.040 | 0.249 ± 0.057 |
>
> - Below, we present the results obtained under different DiFuMo granularity settings. Note: We excluded P \= 64 and 128 from the whole-brain analysis, as these parcellations do not cover the entire brain. Please find the detailed discussion in the corresponding section in Appendix D.14.
>
> | N ROI | FB TCorr | GM TCorr | SC TCorr | CB TCorr | ConnCorr | ConnMSE |
> | :---- | :---- | :---- | :---- | :---- | :---- | :---- |
> | 256 | **0.387 ± 0.059** | **0.415 ± 0.064** | **0.325 ± 0.084** | **0.260 ± 0.074** | **0.591 ± 0.076** | **0.213 ± 0.088** |
> | 512 | 0.367 ± 0.052 | 0.396 ± 0.058 | 0.251 ± 0.037 | 0.247 ± 0.060 | 0.527 ± 0.084 | 0.233 ± 0.072 |
> | 1024 | 0.321 ± 0.036 | 0.358 ± 0.043 | 0.262 ± 0.072 | 0.204 ± 0.058 | 0.485 ± 0.059 | 0.304 ± 0.072 |
>
> We have also included above additional results and discussion in the ***\<Appendix Section D.13, D.14\>*** of the new revised manuscript.
>
>
> >**10. Question 6**
>
> Thank you for your question. Please see our response to this question in our first point ***\< 1. Further zero-shot evaluations on additional public cross-site cohorts and potential future clinical application (W1, Q6) \>***
>
> We again thank the reviewer for the constructive feedback\! We hope our rebuttal addresses your concerns, and please don’t hesitate to let us know if you have any further questions.
>
> **Reference:**
>
> [1] Li, Yamin, et al. "NeuroBOLT: Resting-state EEG-to-fMRI synthesis with multi-dimensional feature mapping." Advances in neural information processing systems 37 (2024): 23378-23405.
> [2] Kovalev, Alexander, Ilia Mikheev, and Alexei Ossadtchi. "fMRI from EEG is only Deep Learning away: the use of interpretable DL to unravel EEG-fMRI relationships." arXiv preprint arXiv:2211.02024 (2022).
> [3] Li, Yamin, et al. "Leveraging sinusoidal representation networks to predict fMRI signals from EEG." Medical Imaging 2024: Image Processing. Vol. 12926\. SPIE, 2024\.
> [4] Hu, Yufan, Wuyang Li, and Yixuan Yuan. "Synthesizing realistic fMRI: a physiological dynamics-driven hierarchical diffusion model for efficient fmri acquisition." The Thirteenth International Conference on Learning Representations. 2025.

---

### Official Review · Reviewer_vkoD · 2025-10-30

**Soundness:** 3
**Presentation:** 3
**Contribution:** 3
**Rating:** 6
**Confidence:** 2

**Summary:**

This paper proposes UniEFS, a unified and context-aware framework for reconstructing full-brain fMRI activity from EEG signals. The key innovation lies in combining self-supervised fMRI pretraining with context-conditioned EEG encoding that incorporates demographic and physiological priors. The model performs frame-wise fMRI reconstruction at the ROI level and achieves strong performance in both resting-state and zero-shot task-based settings.

**Strengths:**

The problem considered in this paper is very interesing. Besides, the idea of injection of auxiliary information to improve the generalization across population is interesting. The experimental validation is solid and convincing.

**Weaknesses:**

The paper utilizes an (\ell_2) loss to enforce alignment between fMRI and EEG. In other works, similarity scores are sometimes used for this purpose. It would strengthen the paper if the authors could provide a comparison between these two approaches. Additionally, the EEG covariance matrix captures neural interactions, which could be leveraged to guide the learning process. However, this aspect is not considered in the current work.

**Questions:**

See the weakness above.

---

> ### Author Response · Authors · 2025-11-27
> **Author  Rebuttal**
>
> We thank the reviewer for their thoughtful and positive assessment of our work. We are glad that the reviewer finds the problem setting compelling and recognizes the novelty of integrating demographic and physiological context to improve population-level generalization. We also appreciate the reviewer’s acknowledgement that our experimental validation is solid and convincing.
>
> We have carefully revised the manuscript according to your suggestions and **added multiple new analyses**. **A new version of the PDF** has been uploaded, with new analysis mainly in ***Appendix*** and with all updated or newly added content highlighted in ***blue*** for ease of review.
>
> Below, we provide detailed, point-by-point responses to each comment.
>
> >**1. Experiment with other latent alignment losses (W1)**
>
> Thank you for the insightful comment. We agree that the choice of alignment loss is important. Here, we have now provided a systematic evaluation of several alignment objectives, including cosine similarity, MSE+cosine, InfoNCE, and contrastive losses. The results are summarized in the table below:
>
> | Loss Type | FB TCorr | GM TCorr | SC TCorr | CB TCorr | ConnCorr | ConnMSE |
> | :---- | :---- | :---- | :---- | :---- | :---- | :---- |
> | MSE (Ours) | **0.367 ± 0.052** | **0.396 ± 0.058** | 0.251 ± 0.037 | **0.247 ± 0.060** | 0.527 ± 0.084 | 0.233 ± 0.072 |
> | InfoNCE | 0.346 ± 0.028 | 0.372 ± 0.017 | **0.270 ± 0.270** | 0.217 ± 0.074 | 0.537 ± 0.167 | **0.180 ± 0.001** |
> | Contrastive | 0.329 ± 0.005 | 0.354 ± 0.007 | 0.256 ± 0.441 | 0.207 ± 0.0184 | 0.516 ± 0.150 | 0.204 ± 0.002 |
> | Cosine | 0.362 ± 0.054 | 0.391 ± 0.063 | 0.266 ± 0.074 | 0.232 ± 0.053 | **0.558 ± 0.091** | 0.212 ± 0.071 |
> | MSE+Cosine | 0.352 ± 0.050 | 0.379 ± 0.076 | 0.256 ± 0.435 | 0.236 ± 0.039 | 0.511 ± 0.153 | 0.246 ± 0.070 |
>
> Specifically, we found that while some alternative losses achieved slightly higher scores on connectivity-related metrics, pure MSE provided the most stable and accurate temporal alignment overall, which is why we selected it as the primary alignment loss. We have added the full results and discussion to the revised manuscript in ***\<Appendix Section D.11, Table 10\>***.
>
> >**2. Experiment with other latent alignment losses (W2)**
>
> We appreciate the reviewer’s insightful comment. We agree that the EEG covariance structure contains meaningful information about neural interactions and could potentially further guide the EEG→fMRI mapping. While our current work focuses on establishing a unified framework with demographic and physiological conditioning directly using raw EEG time series as input, and does not explicitly impose covariance-based constraints, the transformer encoder inherently captures pairwise channel interactions through its self-attention mechanism, which models correlations across channels and time. This enables the model to implicitly encode aspects of EEG covariance during learning.
>
> That said, incorporating *explicit* covariance-aware components, such as (1) connectivity-regularized objectives or (2) channel-interaction priors, represents a very interesting and exciting direction for future research, and we thank the reviewer for highlighting this valuable opportunity.
>
> We again thank the reviewer for the encouraging and constructive feedback. We hope our responses satisfactorily address the concerns, and we would be happy to clarify further if needed.

---

### Official Review · Reviewer_hvue · 2025-10-31

**Soundness:** 2
**Presentation:** 3
**Contribution:** 3
**Rating:** 2
**Confidence:** 4

**Summary:**

This paper proposes UniEFS, a novel transformer-based framework to synthesize full-brain fMRI from low-density EEG recordings, addressing the long-standing challenge of inferring hemodynamic brain activity from electrical scalp measurements. The authors introduce a two-stage training approach leveraging masked fMRI self-supervised pretraining and context-aware conditioning on subject demographics, vigilance state, and dataset identity to enhance EEG-to-fMRI translation. The model is evaluated on multiple resting-state datasets and outperforms previous methods across cortical, subcortical, and cerebellar regions while demonstrating promising zero-shot generalization to task fMRI.

**Strengths:**

- The work pushes boundaries in reconstructing deep brain and full-brain fMRI components from EEG, a problem that is highly relevant for scalable, low-cost brain imaging with broad applications.

- The proposed method integrates pretrained fMRI representations with context-aware EEG encoding, a novel and interesting approach.

- The paper is clearly written and has good presentation

**Weaknesses:**

- The differences reported in the ablation study are small, making it difficult to assess the significance of the contribution of the parts to the model's performance.

- The spatial correlation loss term is not explicitly derived or quantified in analytical detail.

- The proposed method seems to be mostly functional for cortical regions, while performance on sub-cortical regions is poorly explored, diminishing the actual contribution of the model as an effective mapping between EEG and fmri.

- Zero-shot evaluation is limited to one auditory task; broader task validations would strengthen claims of generalizability.

- Some hyperparameter decisions (e.g., 5 dataset tokens) and model architecture lack extensive justification or exploration.

**Questions:**

- In the EEG encoder, the authors use two modules to extract EEG embeddings: a spatio-temporal module and multi-scale spectral transformers. However, the specific contributions of these modules are not clearly assessed. Could you clarify the expected role of each? Is the spatio-temporal module supposed to capture local (fine-grained) features while the multi-scale spectral transformers capture more global or scale-invariant patterns? Do these modules actually complement each other in practice, and is there evidence that both are necessary? It appears that the ablation studies do not test or justify the need for both modules.

- Regarding context tokens, the authors mention using 5 tokens to describe the dataset, with ablation results in Table 2 showing 5 tokens outperform 1 or 10 tokens. However, it’s unclear what exactly these tokens represent or encode. Are they simple dataset identifiers, or do they capture meaningful experimental conditions or metadata? More detail on the nature and creation of these tokens would clarify their role.

- For the baseline models, the authors state that the final projection layer was modified to map embeddings to the selected ROIs. Were these baseline models fine-tuned with the new projection layers? Demonstrating that these models were trained to convergence after modification would strengthen the comparison.

- In the main results, the authors report better prediction performance for cortical networks (somatomotor, dorsal attention, salience/ventral attention), consistent with EEG’s cortical focus. However, subcortical and cerebellar predictions have low temporal correlation (~0.25). This raises a key question: if the model performs well for cortical regions already well represented by EEG, what additional value does this EEG-to-fMRI mapping provide? What new insights does extrapolated fMRI from EEG yield beyond EEG alone? This important question remains inadequately addressed.

- Finally, the ablation studies overall seem to weaken the paper. Most differences are modest and may lack statistical significance, and except for reconstruction loss, other terms contribute marginally. Additionally, the model without EEG-fMRI alignment (i.e., no EEG info) still performs within the standard deviation of the full model. Clarification of these findings and their implications for the model’s robustness and design would be appreciated.

---

> ### Author Response · Authors · 2025-11-27
> **Author Rebuttal - Part 1**
>
> We sincerely appreciate your valuable comments and suggestions, which truly helped us improve the quality of our work. We have carefully revised the manuscript according to your suggestion and **uploaded a new version of the PDF with multiple new analyses.** All updated or newly added content highlighted in ***blue***.
>
> Responses to your specific concerns are presented as follows:
>
> >**1. On the significance of ablation differences (W1, Q5)**
>
> Thank you for your valuable comment. We would like to clarify that the standard deviation in the table reflects variability across 6 evaluation subjects, which naturally captures individual differences in EEG-fMRI alignment. To further quantify the significance of the ablation results, we conducted **paired t-tests** across subjects comparing the full model with the ablated variants (*see revised Table 2 in the updated manuscript*). Although the average numerical improvements appear small, the gains from each module and each loss term are **statistically reliable**. Importantly, we observe **consistent improvements across nearly all subjects**, indicating that the effect is systematic rather than noise-driven and further supporting the contribution of each component in the model.
>
> Regarding the reviewer’s observation that the reconstruction loss produces the larger gain than the alignment loss, we would like to emphasize that although the numerical gain from the alignment loss is smaller, it is **statistically significant and consistently positive across all subjects** (see revised Table 2; metrics *FB TCorr, GM TCorr, Conn PCorr, ConnMSE*). This indicates that the alignment term provides reliable, systematic benefits rather than noise-level variation.
>
> This pattern is consistent with the roles of the two losses in our architecture. In Stage 2, the fMRI decoder is frozen; therefore, the reconstruction loss serves as the primary source of task-specific semantic supervision, guiding the EEG encoder to produce latent representations that remain decodable through the fixed decoder. In contrast, the alignment loss enforces geometric proximity between EEG and fMRI latents but does not account for the decoder’s nonlinear inversion geometry. Because the latent space learned in Stage 1 forms a curved nonlinear manifold, even small off-manifold deviations, though close in Euclidean distance, may decode into semantically incorrect fMRI signals.
>
> As a result, the reconstruction loss naturally yields larger numerical gains, while the alignment loss functions as a **regularizer** that improves latent-space geometry, stability, and consistency, explaining why its improvements are modest in magnitude yet statistically reliable. We appreciate the reviewer pointing this out and have now included this discussion in ***\<Appendix Section E.2\>*** in the revised manuscript.
>
> > **2. Clarification on the spatial correlation loss term (W2)**
>
> Thank you for the question. For each sample (i.e., one fMRI TR), the model predicts a 1x512 vector representing activation across the 512 ROIs. The spatial correlation loss is computed as:  L\_corr(y, y\_hat) \= 1 – corr(y, y\_hat),
> where `corr(y, y_hat)` is the Pearson correlation between the predicted and ground-truth ROI vectors within the same TR. This loss encourages the model to match the spatial activation pattern across ROIs independent of absolute magnitude. We have added this derivation and explanation to *\<Method Section 2.3 EEG-fMRI Embedding Alignment\>* in the revised manuscript.

---

> ### Author Response · Authors · 2025-11-27
> **Author Rebuttal - Part 2**
>
> >**3. On the evaluation and contribution of subcortical regions (W3)**
>
> Thank you very much for the comment. In our original submission, we did evaluate subcortical regions quantitatively and through complementary approaches, including (i) temporal prediction accuracy (SC TCorr) and (ii) subcortical functional connectivity reliability (through FC fingerprinting). In the latter, we specifically tested whether subcortical FC derived from the zero-shot predicted fMRI can preserve individual-specific structure by running subject fingerprinting using only subcortical FC \<Appendix Section D.3, Table 6\>, achieving 90% accuracy compared to 100% when using real fMRI. This demonstrates that, even in subcortical regions, the predicted signals retain meaningful subject-level information. We also provided example subcortical (Putamen, Thalamus) time-series visualizations in the  ***\<Appendix Section D.15, Figure 11\>***.
>
> We would also like to highlight that our work provides, to our knowledge, the first EEG→fMRI model that enables reconstruction of fMRI time courses across the entire subcortical system (along with cortical nodes). Prior studies typically analyze or reconstruct only one or a few subcortical ROIs [1-4], whereas here, we evaluate multiple subcortical regions and multiple metrics (temporal correlation, connectivity reconstruction, and fingerprinting), substantially extending subcortical analysis beyond that of prior work.
>
> We hope this can address your concern, and please let us know if you have further concerns on this.

---

> ### Author Response · Authors · 2025-11-27
> **Author Rebuttal - Part 3**
>
> > **4. Further zero-shot evaluations on broader task (W4) - *PART 1***
>
> We thank the reviewer for raising this important point. To more thoroughly evaluate the model’s zero-shot generalization ability beyond the auditory task included in the paper, we conducted **two additional zero-shot analyses on publicly available datasets**, targeting *both* paired EEG-fMRI settings and real-world EEG-only scenarios. We briefly summarize the setups and findings here; full details and figures have been added to the revised manuscript ***\<Appendix Section D.5. and Figure 7\>***.
>
> **(1) Zero-shot evaluation on an EEG-only Parkinson's disease dataset (real-world no-fMRI scenario)**: To evaluate real-world applicability where fMRI is unavailable \- along with generalizability to a clinical population \- we tested our model on a Parkinson’s EEG dataset [6]  with **task conditions (oddball task) and demographic distribution (elderly participants) entirely different from our training data**.
>
> We segmented the EEG into 16-second windows with a 2-second stride (i.e., 14-second overlap), and fed each window into our trained UniEFS model to extract the ***fMRI-informed EEG latent embedding***. We then compared the t-SNE plots of our model’s embeddings versus the embeddings from the state-of-the-art EEG foundation model CBraMod [7]. ***Please find the new figure in the \<Appendix D.5, Figure 7 of our revised manuscript\>.*** In summary, we have the following key observations:
> * **Clear separation between Parkinson patients and controls**
>    In the zero-shot setting, our latent representations **cleanly separate the PD group from the control group, without any finetuning**. This indicates that the learned EEG→fMRI projection extracts neurophysiologically meaningful structure that generalizes to entirely new populations. Such zero-shot group separation suggests promising potential for downstream clinical applications (e.g., biomarker discovery, screening, or disease monitoring) even when no fMRI is available.
> * **Strong individual-specific clustering**
>   Samples from the same subject cluster tightly together in our latent space, indicating that the EEG→fMRI projection learns highly individualized EEG representations even without explicit subject labels. Incorporating demographic embeddings (age, sex) makes these subject-wise clusters even more compact, suggesting that the model can disentangle individual-specific neural signatures from population-level factors. This pattern is consistent with our fingerprinting results reported in the main paper, where subject identity can be reliably recovered from the predicted fMRI signals. From the latent-space perspective, this further demonstrates that the learned representation is not only discriminative but also structured in a way that preserves stable, subject-specific traits. This also highlights the potential value of our representation for various downstream applications.
> * **Better zero-shot structure than CBraMod**
>   Compared to CBraMod, our embeddings show clearer intra-subject consistency and stronger inter-group separability under a completely zero-shot setting, even though our model is trained on only 28 scans rather than a large-scale EEG corpus. This highlights that the performance does not stem from data scale, but from the inductive bias introduced by learning an fMRI-augmented EEG embedding, where projecting EEG into the semantically structured fMRI latent space provides richer physiological constraints and induces **robust, discriminative, and subject-specific EEG representations**, even in the absence of paired fMRI during inference. This opens the possibility for using fMRI-informed EEG representations as a scalable backbone for future EEG-only downstream tasks.
>
> ***Due to character limits, please find the point (2) in the next response.***

---

> ### Author Response · Authors · 2025-11-27
> **Author Rebuttal - Part 4**
>
> > **4. Further zero-shot evaluations on broader task (W4) - *PART 2***
>
> **(2) Zero-shot evaluation on a public simultaneous EEG–fMRI dataset (motor tasks)**
> We evaluated our model on the **Simultaneous EEG–fMRI Dataset for Multiple Motor Conditions** [5], which contains task paradigms completely different from our data setting. We randomly split the data into training/validation/testing sets with ratio of approximately 2:1:1. Without any retraining or fine-tuning, we directly fed the test set raw EEG into our EEG→fMRI model pretrained on resting-state datasets and assessed the reconstructed fMRI signals.
>
> * As shown in Table below. zero-shot predictions already produce meaningful temporal correlations and connectivity structure, demonstrating stable cross-dataset generalization even when the task paradigm and participant demographics differ substantially from training. Further Finetuning using the training set of the data improves subcortical and cerebellar reconstruction (SC TCorr: 0.158→0.233; CB TCorr: 0.195→0.213) and substantially strengthens connectivity estimation
>
> These analyses together provide additional support that our model generalizes to **novel task structures, new subjects, and unseen EEG–fMRI dynamics**.
>
> | Condition | FB TCorr | GM TCorr | SC TCorr | CB TCorr | ConnCorr | ConnMSE |
> | :---- | :---- | :---- | :---- | :---- | :---- | :---- |
> | Zero-shot | 0.237 ± 0.083 | 0.266 ± 0.087 | 0.158 ± 0.083 | 0.195 ± 0.091 | 0.363 ± 0.114 | 0.257 ± 0.024 |
> | Finetuned | 0.254 ± 0.052 | 0.272 ± 0.057 | 0.233 ± 0.064 | 0.213 ± 0.059 | 0.509  ± 0.060 | 0.139 ± 0.013 |
>
> > **5. Clarification on the dataset tokens  (W5, Q2)**
>
> Thank you for your question. As mentioned in *\<line 232\>* in our original manuscript, a dataset token refers to a learnable embedding that is prepended to the EEG latent tokens before being passed into the transformer. Conceptually, these tokens function similarly to prefix-tuning or prompt-tuning mechanisms: they provide a **lightweight conditioning signal** that allows the model to adapt to dataset-specific characteristics without modifying the main architecture.
>
> Concretely, if the latent dimension is 200, using 1 dataset token means adding a 1×200 learnable embedding, while using 10 tokens corresponds to a 10×200 embedding matrix. For training with multiple datasets, we maintain a small pool of such learnable embeddings, one embedding (of size N×D) per dataset, and **only the token corresponding to the current sample’s dataset is activated** and updated during training, while the others remain untouched. In other words, each sample is associated with a specific dataset token, and only that token is used during the forward and backward pass.
>
> They are not simple dataset IDs. Because they are learned jointly with the rest of encoder and other prompts, they implicitly capture **dataset-level shifts** such as:
> - differences in recording hardware
> - protocol/task differences
> - population differences......
>
> Thus, the dataset tokens serve as a compact embedding that absorbs dataset-specific nuisance factors, allowing the main encoder to focus on modeling subject-level and neural dynamics that generalize across datasets. We found that using **5 dataset tokens** (an embedding size of 5×D) outperforms 1 and 10 under the current dataset scale. This is likely because the number of datasets and the diversity of acquisition conditions in our training setup are relatively limited, so a small token capacity (5 tokens) is sufficient to capture dataset-level shifts without overparameterization.
>
> We also further examined the detaild contribution of all the prompt tokens under a leave-one-site-out setting to see if they mitigate the cross-site domain shift, please see **Figure 8** in ***\<Appendix Section D.6\>*** of our updated manuscript.
>
> At the same time, the dataset token design is fully extensible: if future work involves **more datasets, more heterogeneous recording conditions, or richer metadata**, the number of dataset tokens can be increased (e.g., 10) accordingly to match the complexity of the conditioning space.

---

> ### Author Response · Authors · 2025-11-27
> **Author Rebuttal - Part 5**
>
> > **6. Justification of model architecture (W5, Q1)**
>
> We thank the reviewer for raising this question. Regarding the EEG encoder architecture, our design follows the NeuroBOLT encoder [1] (as mentioned in *line 213* in the original manuscript), which was originally developed and validated for capturing both temporal dynamics and multi-scale spectral structure in EEG. We adopt this architecture *as the encoder backbone* to isolate the contribution of our work, namely, the pretraining strategy, cross-modal alignment, and context conditioning on the encoder side.
>
> The paper [1] has validated the sub-modules of the EEG encoder and demonstrated that the two components in the encoder serve distinct and complementary purposes: (1) **Temporal-Spatial (TS) module:** captures fine-grained temporal structure and local spatial interactions between channels (short-range dynamics). (2) **Multi-Scale Spectral (MSS) module:** captures frequency-specific patterns and scale-invariant structure across spectral bands.
>
> To address the reviewer’s request for empirical evidence, we performed an additional ablation on our task and model setting (i.e., full brain reconstruction, with context-aware conditioning), removing each module individually while keeping the rest of the pipeline unchanged. Results are summarized below (mean ± std across 6 subjects, p<0.05*, p<0.01**, p<0.001***):
>
> | Model Variant | FB TCorr | GM TCorr | SC TCorr | CB TCorr | ConnCorr | ConnMSE |
> | :---- | :---- | :---- | :---- | :---- | :---- | :---- |
> | Ours (Full) | 0.367 ± 0.052 | 0.396 ± 0.058 | 0.251 ± 0.037 | 0.247 ± 0.060 | 0.527 ± 0.084 | 0.233 ± 0.072 |
> | Ours (no MSS) | 0.189 ± 0.076 \*\* | 0.206 ± 0.092 \*\* | 0.126 ± 0.042\*\*\* | 0.139 ± 0.074\*\*\* | 0.122 ± 0.032\*\*\* | 0.407 ± 0.098\*\*\* |
> | Ours (no TS) | 0.350 ± 0.065 \* | 0.376 ± 0.075 | 0.262 ± 0.083 | 0.226 ± 0.055 \*\* | 0.530 ± 0.074 | 0.230 ± 0.080  |
>
> Together, these results demonstrate that the encoder architecture is not arbitrary: the temporal-spatial and multi-scale spectral modules capture **distinct and complementary aspects of EEG**, and **both are empirically necessary** for accurate EEG→fMRI reconstruction. This aligns with the original NeuroBOLT findings and provides direct justification for our architectural choice. We have now included this discussion and table in the revised manuscript ***\<Appendix Section D.12, Table 11\>***.
>
> > **7. Baseline implementation details (Q3)**
>
> Thank you for the question\! We would like to clarify that **all baseline models were trained from scratch**, including the newly added projection layers. This is necessary for a fair comparison because our method is the **only model among all current baselines that performs whole-brain (multi-region) reconstruction**, whereas the original baseline implementations *only support single-region or low-dimensional prediction*. As a result, none of the pretrained weights from the original baseline papers are compatible with our multi-ROI setup.
>
> To make those baselines comparable, we modified each baseline by adding a final projection layer matching our 512-ROI output space, and then **trained the entire model from scratch (or from the pretrain weights for those EEG foundation modell baselines) to convergence under the same training protocol as our model** (optimizer, batch size, number of epochs, learning rate schedule, and early stopping criteria). All baselines reached stable convergence, and we verified that extending training did not yield additional gains.
>
> We have now included these details in the revised manuscript ***\<Appendix C2\>***.

---

> ### Author Response · Authors · 2025-11-27
> **Author Rebuttal - Part 6**
>
> > **8. Why EEG2fMRI mapping provides value beyond EEG alone (Q4)**
>
> We thank the reviewer for this insightful question\! While EEG naturally reflects cortical activity more strongly than subcortical regions, the EEG→fMRI mapping offers several important benefits that extend well beyond what EEG alone can provide.
>
> * **fMRI-augmented latent space yields individualized EEG representations beyond EEG-alone models**
>   A key advantage of the EEG-to-fMRI mapping is that projecting EEG into the structured fMRI latent space injects rich spatial and network-level priors that EEG alone cannot provide. EEG has intrinsically low spatial specificity and cannot separate large-scale networks such as DAN, SN, or SM, nor can it organize signals into ROI-level structure. The fMRI-informed latent space imposes this spatial organization, effectively transforming EEG into a network-aware representation.
>
>   Importantly, this fMRI-augmented representation is not only more structured, it may also be more individualized and robust. As demonstrated in our *zero-shot evaluation on the Parkinson’s disease EEG-only dataset (see response to W4 above)*, the latent embeddings produced by our model: (1) show **clear patient vs. control separation** in a fully zero-shot setting, (2) form **tight subject-specific clusters**, and most importantly, (3) **outperform the state-of-the-art EEG foundation model**, despite our model being trained on far fewer EEG samples. Together, these results show that the EEG→fMRI mapping produces higher-quality, more structured, and more individualized EEG embeddings than EEG-alone training, demonstrating clear value beyond EEG’s native capabilities.
>
> * **Strong performance in regions that EEG alone cannot localize:**
>
>   Many cortical regions where our model performs well \- such as the **insula, dorsal ACC, and other salience-network components** \- are not on the cortical surface and are **well-known to be extremely difficult for EEG source localization**, especially when using only **23 EEG channels** as in our study [8,9].
>   In our original manuscript ***\<Main-text Section 3.4 Figure 4(C)\>***, we provide examples of accurate insula reconstruction and additional qualitative visualizations with successful deep brain region reonstruction (putamen, thalamus) in ***\<Appendix Section D.15, Figure 11\>***. In the revised manuscript ***\<Appendix Figure 6\>***, we also provided the sailence-network time-series reconstruction visualization under zero-shot setting (rest-\>task), which achieves above 0.5 in temporal correlation between real and predicted signals.
>   The fact that our model can accurately reconstruct activation patterns from such regions demonstrates that the EEG→fMRI mapping extracts **structured cross-modal relationships**, rather than merely reflecting superficial EEG potentials.
>
> Overall, while deep-region decoding from EEG is widely recognized as a long-standing and largely unsolved challenge, our results show that meaningful information about subcortical and deeper cortical networks can be extracted through cross-modal modeling. We believe this work opens a promising direction for leveraging population priors and multimodal structure to achieve more comprehensive whole-brain characterization from EEG.
>
>
>
>
> We again genuinely appreciate your constructive suggestions. We truly hope our rebuttal addresses your concerns, and please don't hesitate to let us know if you have any questions!
>
> **Reference:**
> [1] Li, Yamin, et al. "NeuroBOLT: Resting-state EEG-to-fMRI synthesis with multi-dimensional feature mapping." Advances in neural information processing systems 37 (2024): 23378-23405.
> [2] Kovalev, Alexander, Ilia Mikheev, and Alexei Ossadtchi. "fMRI from EEG is only Deep Learning away: the use of interpretable DL to unravel EEG-fMRI relationships." arXiv preprint arXiv:2211.02024 (2022).
> [3] Li, Yamin, et al. "Leveraging sinusoidal representation networks to predict fMRI signals from EEG." Medical Imaging 2024: Image Processing. Vol. 12926\. SPIE, 2024\.
> [4] Meir-Hasson, Yehudit, et al. "An EEG finger-print of fMRI deep regional activation." Neuroimage 102 (2014): 128-141.
> [5] Bondi, Elena, et al. "Investigating the neurovascular coupling across multiple motor execution and imagery conditions: a whole-brain EEG-informed fMRI analysis." NeuroImage (2025): 121311\.
> [6] Cavanagh, J. F. "EEG: 3-Stim auditory oddball and rest in Parkinson’s." OpenNeuro, OpenNeuro (2021).
> [7] Wang, Jiquan, et al. "CBraMod: A Criss-Cross Brain Foundation Model for EEG Decoding." The Thirteenth International Conference on Learning Representations.
> [8] Iachim, Evelina, et al. "Automated electrical source imaging with scalp EEG to define the insular irritative zone: comparison with simultaneous intracranial EEG." Clinical Neurophysiology 132.12 (2021): 2965-2978.
> [9] Michel, Christoph M., et al. "EEG source imaging." Clinical neurophysiology 115.10 (2004): 2195-2222.

---

### Official Review · Reviewer_rAYg · 2025-10-31

**Soundness:** 3
**Presentation:** 3
**Contribution:** 3
**Rating:** 4
**Confidence:** 4

**Summary:**

The paper proposes UniEFS, a unified model that reconstructs fMRI signals from EEG. The method combines a pretrained fMRI decoder (learned via masked modeling) with a context-aware EEG encoder that includes demographic and vigilance tokens. The approach aims to handle cross-subject variability and achieve full-brain reconstruction from EEG using a single model.

**Strengths:**

•	Ambitious and well-motivated approach tackling a challenging multimodal translation problem.
•	The two-stage framework (fMRI pretraining + EEG-fMRI alignment) is conceptually sound and leverages large-scale unpaired data effectively.
•	Comprehensive experiments showing improvement over prior EEG-to-fMRI synthesis models.
•	Context conditioning via metadata (age, sex, vigilance) is innovative and biologically meaningful.

**Weaknesses:**

•	Potential data leakage concern: the pretrained fMRI decoder is later fine-tuned using data that overlaps with the EEG-fMRI pairs used in the alignment stage. This could inflate performance metrics. The paper should clarify whether fMRI data used for both LoRA finetuning and EEG-to-fMRI translation training step were fully disjoint from evaluation subjects.
•	The alignment loss seems to have marginal benefit according to the ablation table. The authors should explain whether alternative alignment methods (e.g., contrastive or InfoNCE-style objectives) were tested and why simple MSE was chosen.
•	The model is trained for next-frame prediction, which assumes future fMRI frames depend only on preceding EEG. However, since the EEG window already includes pre-fMRI dynamics, it’s not clear whether the model truly predicts unseen future frames or partially reconstructs signals already reflected in the EEG.

**Questions:**

1.	Can you clarify whether fMRI data used for decoder pretraining or fine-tuning overlaps with EEG-fMRI training subjects? If so, this could lead to data leakage.
2.	Have you tried contrastive alignment or other objectives (e.g., cosine similarity, CCA, InfoNCE)? If tested, what were the differences in alignment quality or downstream reconstruction?
3.	The ablation shows the alignment loss doesn’t contribute much—can you elaborate on why it might be?
4.	Given the temporal nature of EEG and fMRI, did you test predicting more than one frame ahead, or is the model implicitly reconstructing known signals? Also have to try multiple timescale for forward prediction? Any of them produce a benefit?
5.     The vigilance embedding results are interesting—would be nice to see whether similar gains appear for task-based fMRI.

---

> ### Author Response · Authors · 2025-11-27
> **Author Rebuttal - Part 1**
>
> We sincerely thank the reviewer for the thoughtful review and constructive feedback\! We are delighted that the reviewer recognized the ambition and motivation of our approach, the soundness of our novel framework and methodology, the comprehensiveness of our experimental evaluation, and the biological relevance of our context-aware design.
>
> We have carefully revised the manuscript according to your suggestion and **uploaded a new version of the PDF**, with all updated or newly added content highlighted in ***blue***. Please find our point-by-point responses to your questions below:
>
> > **1. Clarification on Finetuning on fMRI on paired dataset (W1, Q1)**
>
> Thank you for raising this point. We would like to clarify that there is no data leakage in any stage of our framework.
> - **Decoder fine-tuning uses only the training subjects.**
> During the f-MSM fine-tuning stage, we use only the fMRI scans from the **training split**, with train/validation/test subjects strictly disjoint. The model is fine-tuned for 20 epochs, and the checkpoint is selected **exclusively based on validation loss**, without accessing any fMRI data from test subjects at any point. We explicitly added this clarification to the manuscript to avoid potential ambiguity. The purpose of this fine-tuning step is solely to mitigate site- or device-related domain shifts between the pretraining dataset and the target dataset, and does not involve any information leakage from evaluation subjects.
>
> - **EEG-fMRI alignment stage also uses only training subjects.**
> Similarly, the EEG-fMRI alignment model is trained only on training subjects, and the alignment stage does not access any fMRI data from test subjects either. No information (EEG, fMRI, or metadata) from evaluation subjects is used at any point in the training pipeline. All evaluations are strictly on held-out subjects.
>
> - **Finetuning provides only marginal improvements, confirming the absence of leakage.**
> As shown in Table 2 of the original manuscript, removing fine-tuning yields almost identical performance across metrics. Fine-tuning led to only a small improvement in the connectivity reconstruction metric (ConnMSE, from 0.27 to 0.23; lower is better), while temporal-correlation-based metrics (Tcorr, ConnCorr) remain stable. This further indicates that fine-tuning is not exploiting any leakage but simply helps adjust for domain differences between datasets.
>
>
> > **2. Experiment with other latent alignment losses (W2, Q2)**
>
> Thank you for the insightful comment. We agree that the choice of alignment loss is important, and we would like to clarify that our use of MSE was not arbitrary. During model development, we systematically evaluated several alignment objectives, including cosine similarity, MSE+cosine, InfoNCE, and contrastive losses. We found that while some alternative losses achieved slightly higher scores on connectivity-related metrics, pure MSE provided the most stable and accurate temporal alignment overall, which is why we selected it as the primary alignment loss. For brevity, we initially reported only the main configuration in the manuscript. For completeness, we now include the full comparison table in both the revised manuscript and below:
>
> | Loss Type | FB TCorr | GM TCorr | SC TCorr | CB TCorr | ConnCorr | ConnMSE |
> | :---- | :---- | :---- | :---- | :---- | :---- | :---- |
> | MSE (Ours) | **0.367 ± 0.052** | **0.396 ± 0.058** | 0.251 ± 0.037 | **0.247 ± 0.060** | 0.527 ± 0.084 | 0.233 ± 0.072 |
> | InfoNCE | 0.346 ± 0.028 | 0.372 ± 0.017 | **0.270 ± 0.270** | 0.217 ± 0.074 | 0.537 ± 0.167 | **0.180 ± 0.001** |
> | Contrastive | 0.329 ± 0.005 | 0.354 ± 0.007 | 0.256 ± 0.441 | 0.207 ± 0.018 | 0.516 ± 0.150 | 0.204 ± 0.002 |
> | Cosine | 0.362 ± 0.054 | 0.391 ± 0.063 | 0.266 ± 0.074 | 0.232 ± 0.053 | **0.558 ± 0.091** | 0.212 ± 0.071 |
> | MSE+Cosine | 0.352 ± 0.050 | 0.379 ± 0.076 | 0.256 ± 0.435 | 0.236 ± 0.039 | 0.511 ± 0.153 | 0.246 ± 0.070 |
>
> We have also added the above results and discussion to the revised manuscript  ***\<Appendix Section D.11, Table 10\>***

---

> ### Author Response · Authors · 2025-11-27
> **Author Rebuttal - Part 2**
>
> > **3. Clarification on the training objective (W3, Q4)**
>
> Thank you for raising this important point. We would like to clarify that our model is *not* designed as a future-frame forecasting model. Our objective is to reconstruct the hemodynamic (fMRI) response that corresponds to the neural activity contained in the preceding EEG window, rather than predicting genuinely unseen future fMRI states.
>
> Physiologically, the BOLD signal reflects neural activity with a slow hemodynamic delay (on the order of several seconds). Therefore, the fMRI frame at time *t* is largely driven by neural dynamics occurring in the preceding EEG window. Our formulation leverages this well-established temporal coupling, **mapping EEG-derived neural activity into the corresponding BOLD representation.**
>
> Regarding the reviewer’s question about multi-step or further-ahead prediction: we did not evaluate multi-frame forecasting or multi-timescale prediction in this work. Extending our framework to genuine forecasting would require carefully isolating the effect of fMRI autocorrelation and ensuring there is no inadvertent information leakage from temporally adjacent frames. This represents a conceptually different problem setting—one that we agree is interesting and important, but outside the scope of our current reconstruction-focused study. We thank the reviewer for highlighting this distinction, and we will clarify this more explicitly in the revised manuscript ***\<Appendix Section E.3\>***.
>
>
> > **4. Contribution of the alignment loss and discussion of the performance (Q3)**
>
> We thank the reviewer for raising this point. Indeed, we see that the reconstruction loss contributes the most to overall performance, while the alignment loss provides a smaller numerical improvement. However, we think this behavior is expected given our two-stage design. In Stage 2, the fMRI decoder is frozen, and thus the reconstruction loss supplies the only task-specific semantic supervision: its gradients propagate through the fixed decoder’s Jacobian, guiding the EEG encoder to produce latent representations that the decoder can reliably map back into valid fMRI signals. In contrast, the alignment loss enforces only geometric proximity between EEG and fMRI latents without considering the decoder’s nonlinear inversion geometry. Because the latent space learned in Stage 1 forms a nonlinear manifold, even small off-manifold deviations, though close in L2 distance, may decode into semantically incorrect fMRI outputs. Therefore, the reconstruction loss plays the dominant role in ensuring that EEG latents remain “decodable,” whereas the alignment loss acts primarily as a regularizer that improves latent-space geometry and cross-subject consistency.
>
> Importantly, the improvement from alignment loss is **statistically significant across subjects**.  Below, we conducted paired t-tests between the full model and the variant without the alignment loss.
>
> | Model Type | FB Tcorr | GM Tcorr | Conn PCorr | Conn MSE |
> | :---- | :---- | :---- | :---- | :---- |
> | Full | 0.367 ± 0.052\*\* | 0.394 ± 0.060\* | 0.527 ± 0.084\*\* | 0.233 ± 0.072\*\* |
> | w/o L\_align | 0.339 ± 0.052 | 0.367 ± 0.055 | 0.502 ± 0.082 | 0.280 ± 0.083 |
>
> Significance：p\<0.05 (\**), p\<0.01 (*\*\*), p\<0.001(\*\*\*), uncorrected.
>
> These results show that although the magnitude of improvement is smaller than that of the reconstruction loss, the gain from the alignment loss is statistically reliable, reflecting **consistent benefits across nearly all subjects**. This supports our interpretation that the alignment loss enhances latent geometry and stability, complementing the stronger reconstruction objective.

---

> ### Author Response · Authors · 2025-11-27
> **Author Rebuttal - Part 3**
>
> > **5. Gains of vigilance tokens in zero-shot setting (Q5)**
>
> Thank you for this valuable suggestion\! We have updated Figure 3 in the revised manuscript to include visualizations of the vigilance-token gains in the zero-shot setting (rest → auditory task), highlighting both the most improved cortical regions and their corresponding network assignments. Overall, two networks show the most pronounced improvements.
>
> **First**, the *Temporo-Parietal (TempPar)* network exhibits the largest performance gain when incorporating the vigilance token. This network is known to support attentional reorienting, sensory–motor integration, and response preparation \- functions that are strongly modulated by moment-to-moment vigilance and arousal. In the auditory task dataset, participants must detect auditory cues and make rapid button-press responses; thus, fluctuations in alertness directly impact both auditory processing efficiency and motor readiness, processes for which TempPar plays a central role.
>
> **Second**, the *Dorsal Attention Network (DAN)* also shows substantial improvements with vigilance conditioning. DAN is one of the networks most sensitive to arousal and sustained attention. Because our zero-shot auditory task relies heavily on top-down attention and continuous task engagement, incorporating a vigilance token provides critical state information that enables more accurate reconstruction of DAN-related dynamics.
>
> As shown in the quantitative results below, incorporating the vigilance token consistently improves the reconstruction performance.
>
> | Zero-shot Setting | Model | FB TCorr | GM TCorr | SC TCorr | CB TCorr | ConnCorr |
> | :---- | :---- | :---- | :---- | :---- | :---- | :---- |
> | With Vig | **0.354 ± 0.058** | **0.369 ± 0.062** | **0.294 ± 0.088** | **0.299 ± 0.052** | **0.532 ± 0.108** | **0.265 ± 0.064** |
> | No Vig | 0.312 ± 0.048 | 0.331 ± 0.036 | 0.284 ± 0.101 | 0.298 ± 0.083 | 0.512 ± 0.052 | 0.311 ± 0.046 |
>
>
> We again thank reviewer for the constructive and valuable feedback! We hope our rebuttal address your concerns, and please let us know if you have further questions!

---

### Author Response · Authors · 2025-12-03
**Review and Author Rebuttal Summary (1/3)**

Dear PCs, SACs, ACs, and Reviewers,

We sincerely appreciate the time and effort the reviewers have devoted to evaluating our manuscript. We are encouraged by the positive feedback and grateful for the recognition of our work’s contributions to the field. In particular, we appreciate the reviewers’ acknowledgment of:

* **Ambition & Novel Formulation:**
  Reviewers agreed on the ambition and novelty of tackling unified, frame-wise whole-brain EEG-to-fMRI synthesis: a challenging multimodal translation task of high scientific relevance and broad application. They described our formulation as *“ambitious and well-motivated”* (***rAYg***), *“pushing boundaries in reconstructing deep and full-brain components”* (***hvue***), and *“very interesting”* (***vkoD***), underscoring the originality and relevance of our problem setup.
* **Architecture & Methodological Soundness:**
  The proposed two-stage framework (masked fMRI pretraining followed by EEG–fMRI latent alignment) was recognized as ***conceptually sound*** and an **effective way to leverage large-scale unpaired data** (***rAYg; hvue***). Reviewers also noted the value of injecting auxiliary demographic and vigilance cues, characterizing our context-aware design as *“innovative and biologically meaningful”* (***rAYg***), *“novel and interesting”* (***hvue***), and *“an interesting idea for improving generalization across population”* (***vkoD***).
* **Strong Empirical Validation:**
  Reviewers pointed to *“solid and convincing experimental validation”* (***vkoD***), *“comprehensive experiments showing improvement over prior EEG-to-fMRI models”* (***rAYg***), and *“solid baselines and component ablations”* (***h7XH***). Demonstrated improvements across cortical, subcortical, and cerebellar regions were viewed as impactful and meaningful for scalable neuroimaging.
* **Biological Plausibility & Interpretability:**
  Reviewers appreciated the biological grounding of the method, noting that our metadata conditioning (age, sex, vigilance) aligns with known physiological variability (***rAYg***) and that network-wise effects, especially vigilance-related modulation, are *“compelling and biologically plausible”* (***h7XH***). This supports the neuroscientific validity of our approach.
* **Clarity & Presentation Quality:**
  Comments highlighted that the paper is *“clearly written and has good presentation”* (***hvue***) and *“structured and motivated with clear training pipeline and evaluation”* (***h7XH***). Figures and explanations were noted as effective and accessible.

---


We have carefully reviewed all comments and conducted extensive additional experiments and analyses in response. All major revisions in the updated manuscript are highlighted in ***blue***. A summary of the key improvements is provided below:

* **Justifying model components and validating the contribution of each design choice** (rAYg: Weakness 2; hvue: Weakness 1, Question 1, 5; vkoD, Weakness 1; h7XH: Question 3):
  **Our response:**
  * We conducted **comprehensive paired t-tests between the proposed model and every ablated variant**, demonstrating that each component \- particularly the **alignment loss**, along with pretraining, prompt tokens, and encoder modules \- yields reliable and statistically significant improvements. *These results reflect consistent gains of our model/loss design across nearly all subjects* (**Main Text, Table 2, Page 9; Appendix D.12, Page 26; Appendix E.2, Page 27**).
  * Additionally, we **conducted extensive leave-one-site-out zero-shot and within-site evaluations with full prompt-type ablations** (dataset, age, sex, vigilance, pretraining), *demonstrating the model’s OOD (out-of-distribution) generalizability and showing that each prompt consistently improves cross-site performance* **(Appendix D.6, Figure 8, Page 22-23).**
  * To justify the choice of MSE for latent alignment, **we compared MSE, cosine, contrastive, InfoNCE, and MSE+cosine losses**. MSE provides the most stable and accurate temporal alignment, and adding the alignment loss yields consistent, statistically significant gains across subjects. *This resolves questions about loss choice and its benefit.* **(Appendix D.11, Page 25-26)**
  * Together, these analyses strengthen the justification of our architectural design and directly address the reviewers’ requests for statistical rigor and clearer ablation interpretation.

---

> ### Author Response · Authors · 2025-12-03
> **Review and Author Rebuttal Summary (2/3)**
>
> * **Points regarding external validity, cross-task generalization beyond resting-state, and potential clinical applicability.** (hvue: Weakness 4, h7XH: Weakness 1\)
>   **Our response:**
>   * We added **two substantial new zero-shot evaluations on two new datasets with unseen task conditions**: (a) A Parkinson’s disease EEG-only clinical dataset (no fMRI), demonstrating **clearer patient–control separation and stronger subject-specific structure** in latent space **compared with the state-of-the-art EEG foundational model**, *highlighting the potential clinical application, such as biomarker discovery and disease monitoring*. (b) A public simultaneous EEG-fMRI motor-task dataset, *showing meaningful zero-shot performance and improved results after optional fine-tuning.* (**Appendix D.5, Page 20-22, Figure 7**)
>
> * **Physiological validity and network-level characterization of generated fMRI.** (h7XH: Weakness 4\)
>   **Our response:**
>   * As suggested by Reviewer rAYg (Question 5), we further analyzed the **whole-brain effects of vigilance tokens in the zero-shot setting** and provided **additional visualizations and discussion**. Our analysis shows that vigilance conditioning yields the strongest gains in the Temporo-Parietal and Dorsal Attention networks during zero-shot prediction in the auditory task, two systems tightly linked to arousal, attentional control, and task engagement, *highlighting the neuroscientific relevance of the vigilance token* **(Main text, Figure 3, Page 8-9)**.
>   * As suggested by Reviewer h7XH, we included a network-wise temporal dynamics analysis in the revised manuscript. We averaged the ROI time series to obtain a network-level signal and computed the temporal correlation between predicted and ground-truth activity. *The resulting average TCorr of \~0.5 demonstrates strong network-level consistency in the reconstructed fMRI*. **(Appendix D.4, Figure 6, Page 20-21)**
>   * As suggested by Reviewer h7XH, we added a new spectral evaluation of the reconstructed fMRI signals. The predicted and true PSDs exhibit very high correspondence across frequencies (Pearson r \= 0.93, p \< 0.001; Fig. 9(B)), *demonstrating that the model captures the correct temporal dynamics underlying fMRI fluctuations.* **(Appendix Section D.7, Figure 9, Page 23-24)**
>
> * **Concerns about potential data leakage during decoder fine-tuning and alignment.** (rAYg: Question 1; h7XH: Weakness 3\)
>   **Our response:**
>   * We clarified that all fine-tuning and alignment stages use only training subjects, with strictly disjoint train/val/test splits. No fMRI, EEG, or metadata from evaluation subjects is seen during any training step. We further show that removing fine-tuning yields nearly identical performance, confirming the absence of leakage. **(Main text, Table 2, Page 9\)**
>
> * **Evaluation of deep cortical/subcortical regions and added value beyond EEG-only representations** (hvue: Weakness 3, Question 4\)
>
> 	**Our response:**
>
>   * We clarified and highlighted that the predicted subcortical signals retain meaningful individual-specific structure with *strong performance in FC fingerprinting accuracy.* **(Appendix D.3, D.15, Page 20, 26-27).**
>   * We further strengthened biological plausibility analyses, showing **accurate zero-shot reconstruction** in regions that EEG struggles to localize (insula, dorsal ACC, salience-network components, putamen, thalamus) and *robust Yeo-17 network dynamics (average TCorr ≈ 0.5), including zero-shot salience-network reconstruction* **(Appendix D.4, D.15, Page 20-21, 26-27).**
>   * Finally, zero-shot evaluation on a Parkinson’s EEG-only dataset shows that our fMRI-informed latent space yields clearer patient–control separation and tighter subject-specific clustering than an EEG-only foundation model, *demonstrating the added representational value introduced by EEG→fMRI mapping* **(Appendix D.5, Page 20-22)**.
>
> * **Exploratory analyses on temporal context and spatial granularity.** (h7XH, Question 5\)
>   **Our response:**
>   * Following Reviewer h7XH’s suggestion, we added several exploratory analyses to better understand how temporal context and spatial granularity interact with model performance. These include **window-length ablations** (4s/8s/12s/16s, with 16s showing the strongest TCorr) presented in **Appendix D.13 (Page 26\)**, **parcellation-scale explorations** (256/512/1024 ROIs) shown in **Appendix D.14 (Pages 26–27)** to examine resolution–stability trade-offs, and **frequency-domain perturbation tests** included in our **direct response to Reviewer h7XH**. *These exploratory results provide additional insight into the model’s behavior under different temporal and spatial configurations.*

---

> > ### Author Response · Authors · 2025-12-03
> > **Review and Author Rebuttal Summary (3/3) - Thank you!**
> >
> > Above, we have summarized the **key additional analyses and revisions** introduced during the discussion period, with the hope that this assists the AC in their evaluation. **Many further supporting experiments and clarifications, including smaller or more detailed analyses, are documented in our point-by-point rebuttal and the revised manuscript.**
> >
> > We are sincerely grateful to the reviewers, AC, SAC, and PC, for their time, expertise, and thoughtful feedback. Their invaluable insights have significantly strengthened the paper, and we offer our deepest appreciation to everyone involved.

---

### Meta-Review · Area_Chair_9EGs · 2026-01-09

**Summary:**

This submission presents UniEFS, a unified and context-aware framework for EEG-to-fMRI reconstruction. Reviewers acknowledged the work’s methodological ambition but raised critical concerns about weak ablation studies and unstable performance across comparisons. While the authors supplemented statistical tests (paired t-tests) and expanded zero-shot evaluations to address these issues, the core limitation of unstable ablation effects remains unresolved, likely due to the limited number of test subjects, which undermines the robustness of the conclusions.

**Reviewer Concerns:**

**Addressed concerns:** Data leakage, cross-task generalization, and physiological validity.

**Outstanding concerns:** Insufficiently robust ablations and small dataset, which are core flaws that undermine the reliability of the study’s conclusions. Statistical significance alone cannot offset the instability introduced by small sample sizes and inconsistent trends.

**Reviewer Scores:**

Reviewer rAYg (Rating: 4 → Adjusted: 4): Paired t-tests may not completely eliminate reviewers' concerns about subtle differences among ablations.

Reviewer hvue (Rating: 2 → Adjusted: 4): Subcortical performance and generalization were partially improved via supplementary analyses, but unreliable ablation was not completely resolved.

Reviewer vkoD (Rating: 6 → Adjusted: 6): Concerns solved.

Reviewer h7XH (Rating: 6 → Adjusted: 5): Concerns are almost solved, but erratic ablation effects and small dataset still weaken the study’s rigor.

---

### Decision · Program_Chairs · 2026-01-26

Reject